# LEGO-COMPILER: ENHANCING NEURAL COMPILATION THROUGH COMPOSABLE CHAIN OF THOUGHT

## ABSTRACT

Large language models (LLMs) have the potential to revolutionize how we design and implement compilers and code translation tools. However, existing LLMs struggle to handle long and complex programs. We introduce LEGO-Compiler, a novel neural compilation system that leverages LLMs to translate high-level languages into assembly code. Our approach centers on three key innovations: LEGO translation, which decomposes the input program into manageable blocks; annotation-based Chain-of-Thoughts, guiding LLMs through the compilation process with LLM-annotated context; and a feedback mechanism for self-correction. Supported by formal proofs of code composability, LEGO-Compiler demonstrates high accuracy on multiple datasets, including over 99% on ExeBench and 100% on industrial-grade CoreMark, and successfully handles programs far exceeding the length limitations of native LLM translation. This work opens new avenues for applying LLMs to system-level tasks, complementing traditional compiler technologies.

## 1 INTRODUCTION

The rapid development of Large Language Models (LLMs) has led to an expansion of their applications and effectiveness across various domains (Rombach et al., 2022; OpenAI, 2023; 2024; Ziegler et al., 2024). One important area where LLMs have shown impressive results is code translation, including tasks such as code generation from natural languages (Zan et al., 2023) and transformation between programming languages (Yang et al., 2024). In code translation, LLMs have demonstrated remarkable accuracy and readability, often surpassing manually crafted translators.

While LLMs have shown promising results in translating between high-level programming languages (Rozière et al., 2020; Roziere et al., 2021; Szafraniec et al., 2023) and in decompilation tasks (Fu et al., 2019; Cao et al., 2022; Armengol-Estapé et al., 2023), their application to translating from high-level languages to low-level assembly languages remains a relatively unexplored area. This can be attributed to two main factors. Firstly, the dominance of traditional compilers in this domain has left little incentive for exploring alternatives in such a mature field. Secondly, the complexity of compilers and the bitwise precision required in compilation tasks for semantic accuracy have made it challenging for LLMs based on statistical learning.

Despite these challenges, LLMs have shown promising capabilities in compilation-related tasks. Cummins et al. (2023; 2024) has demonstrated their proficiency in optimizing compiler options and their excellent ability to mimic compiler code behavior, producing high-quality IR code. Furthermore, preliminary explorations in translating from high-level languages to assembly languages, such as C-x86 (Armengol-Estapé & O'Boyle, 2021) and C-LLVM IR (Guo & Moses, 2022), have indicated the potential feasibility of using LLMs in compilation tasks. However, these existing works have not fully addressed the boundaries of LLM capabilities in compilation tasks – specifically, what LLMs can and cannot do in this domain. Compilation is typically divided into two main aspects: functionality and optimization. This work focuses on exploring and answering questions about LLM capabilities in the functionality aspect of compilation.

LLMs are pre-trained on vastly large code corpora. some are monolingual, and some may be bilingual (where LLMs can learn the translation rule between two languages). However, most of these LLMs do not disclose their training datasets, so their capabilities can only be assessed through

empirical testing. We primarily find that LLMs learn the neural compilation process from directly compiler-generated bilingual corpora, which is a relatively easy way to construct pretraining dataset. However, we found that assembly code directly generated by traditional compilers is hard to learn for LLM-based generation due to several challenges. These include the presence of semantically opaque labels, symbols or numeric values that LLMs struggle to translate accurately, and the need to handle symbol renaming for identifiers with the same name in different scopes, etc. Although style migration or modifications to existing compilers can be made, these approaches still rely on an existing compiler to perform the neural compilation job, which doesn't outperform existing designs.

Our work takes a different approach where we do not require bilingual corpora. As a result, we don't rely on an existing compiler. Regarding LLMs' strong in-context learning abilities (Min et al., 2022; Song et al., 2024), we propose the following methods: through high-quality examples and compiler knowledge guided Chain-of-Thoughts, LLMs can perform step-by-step neural compilation. This approach involves generating annotations highly corresponding to source code statements and data structure layout annotations, leading to substantial improvements in the compilation generation task.

More importantly, the scalability of current code translation is also a big problem. Although advanced LLMs already have hundreds of thousands tokens context limit, they can not merely compile a code with 2.6k tokens in CoreMark (Gal-On & Levy, 2012), which is just a 200-LOC function. To address this limitation, we have an intuitive thought: can we split the program into finer-grained components, compile each component, then assemble them together? We can surely do it in function-level, since functions are trivial semantic units (Ibrahimzada, 2024). However, we still think function-level is still too coarse-grained, and we seek to further break it.

Based on these insights, we propose a novel approach called LEGO translation, which draws inspiration from the modular and composable nature of LEGO blocks. This method breaks down large programs into manageable, semantically-composable control blocks, analogous to LEGO pieces. These blocks are then independently translated and rebuilt to form a much larger scale translation. We apply the LEGO translation method to the compilation domain and, guided by a series of compilation-specific expert knowledge, design the LEGO-Compiler, a scalable, LLM-driven system that leverages the power of LLMs to perform neural compilation tasks.

LEGO-Compiler can correctly compile over 99% of the code in ExeBench (Armengol-Estapé et al., 2022), a large scale dataset through careful unit-testing. We can also correctly compile 100% Core-Mark (Gal-On & Levy, 2012), an industrial-grade code that encompasses most common programming language features in C. Regarding scalability, we have verified that LEGO translation method can significantly scale up the capability of code translation/compilation performed by LLMs, where we propose the LongFunction dataset for very long code translation and compilation evaluation, and the LEGO translation method can sufficiently translate it for both neural compilation or code translation purposes.

The main contributions of this work are as follows:

- We propose a set of novel methods for neural compilation, including LEGO translation for breaking down large programs, annotation-based Chain-of-Thoughts (CoTs) that explicitly generate intermediate results to aid translation, and a feedback-driven self-correction mechanism. These methods collectively address the challenges of applying LLMs to complex compilation tasks.

- We introduce LEGO-Compiler, a comprehensive neural compilation system that integrates our proposed methods. LEGO-Compiler incorporates control flow annotation, struct annotation, and variable mapping to ensure accurate and scalable compilation across various architectures and programming languages.

- We provide both theoretical and empirical support for our approach. We present a formal proof of code composability that underpins the LEGO translation method. Empirically, we demonstrate LEGO-Compiler's effectiveness through extensive evaluations, achieving over 99% accuracy on ExeBench and 100% accuracy on the industrial-grade CoreMark benchmark. Our system successfully handles programs far exceeding the length limitations of direct LLM translation, showcasing its scalability.

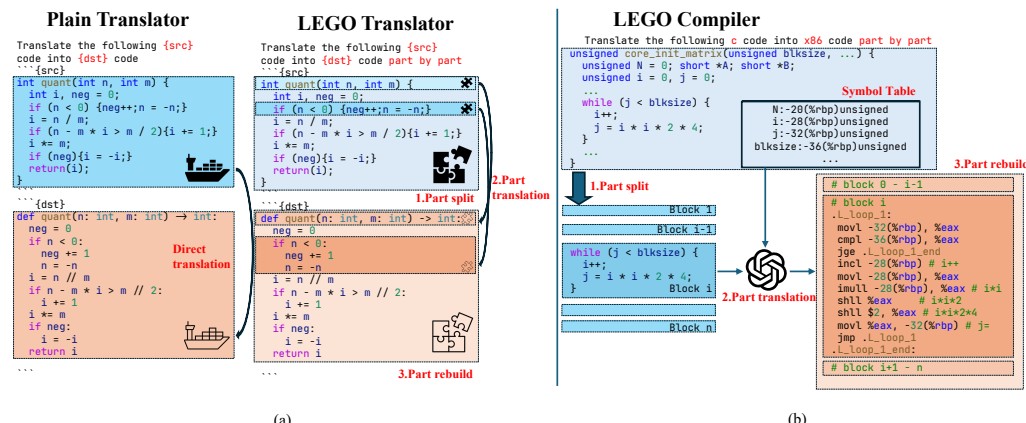

Figure 1: **a.** Plain translation vs LEGO translation, by splitting the program into smaller composable control blocks(parts), translating each part becomes an easier task, and rebuilding each translated partial result will form a full translation. **b.** LEGO compiler, a special case for LEGO translation, to translate each part correctly, a symbol table need to be maintained first and provided during translation.

## 2 METHODS

### 2.1 PROBLEM DEFINITION

Before introducing our method, we first define the neural compilation problem. Neural compilation can be viewed as a specialized version of code translation problem, as defined in Definition 1, with the goal of translating high-level programming language as the *src* language (such as C) into low-level assembly language as the *dst* language (such as x86, ARM, or RISC-V). Unlike general code translation, compilation needs to handle more low-level details, such as memory layout and calling convention, while ensuring the functional correctness of the translated result.

**Definition 1.** *There are two programming languages: $\mathcal{L}_{src}$ and $\mathcal{L}_{dst}$, each is an infinite set of valid program strings. There exists a unary relation $\rightharpoonup$ from $\mathcal{L}_{src}$ to $\mathcal{L}_{dst}$. The problem is to perform a translator function T: $\forall x \in \mathcal{L}_{src}, (\exists u \in \mathcal{L}_{dst}, x \rightharpoonup u) \rightarrow (x \rightharpoonup T(x)), T(x) \equiv x$ semantically.*

### 2.2 **LEGO TRANSLATION**: CORE METHOD

As depicted in (a) in Figure 1, previous neural code translation methods typically convert entire programs at the function or file level. While this approach may be effective for smaller programs, it struggles with larger programs due to significant accuracy degradation. These methods translate code at a coarse granularity, making it challenging to translate very long functions using LLMs. Taking neural compilation as an example, all current LLMs fail to compile a C function with larger than 2.6k tokens using direct translation, although some advanced LLMs already have 128k-200k context limit. They could also perform code-snippet level translation, but they lack guidelines and necessary information to compose the code-snippet level results together, and there is also no clear formal proof to the composability of code. Despite these limitations, we observe an inherently composable nature in code. In the context of neural compilation, we propose the following insights to enhance translation scalability:

- Fine-grained translation: Instead of translating an entire program at once, focus on translating smaller code snippets accurately. By ensuring each part is correctly translated, they can be combined to form a semantically equivalent complete translation.

- Contextual Awareness: Effective translation of smaller code snippets requires understanding their contextual positioning within the code. This includes recognizing the relationship with preceding and succeeding snippets to maintain semantic coherence.

- Symbol Handling: Accurate translation involves reasoning about necessary symbols and constructs within each snippet to enhance alignment with the target language's syntax and semantics, thereby aiming to preserve the intended functionality.

Inspired by Wang et al. (2024), where this process is similar to the destruct and rebuild process of a LEGO toy, we named the fine-grained translation technique as **LEGO translation** and our system built upon it as **LEGO-Compiler**. As depicted in (b) in Figure 1, LEGO translation first breaks down large programs into manageable, self-contained blocks, analogous to LEGO pieces (**Part split**). Then these blocks are independently translated (**Part translation**) and finally recombined, enabling scalable and accurate translation of complex programs (**Part rebuild**). All these methods rely on an inherently nature in programming languages, the composability in control block level, which reflects the linearization process in compiler design (Wirth et al., 1996), where tree-structured control flow can be linearized, and therefore, composable. We have formally proved the widely applicable composability of programming languages using a constructive approach in Appendix B.

---

**Algorithm 1** LLM-driven **Part Split** Algorithm based on Control Blocks

---

   **procedure** SELECTCONTROLBLOCKS($function$)
      $blocks \leftarrow \emptyset$
      $deque.push\_back(function)$
      **while** $deque$ is not empty **do**
         $block \leftarrow deque.pop\_front()$
         $decision \leftarrow$ LLMDecideSplit($block$)
         **if** $decision$ is "keep" **then**
            $blocks$.append($block$)
         **else**
            $subBlocks \leftarrow$ SplitByOutermostControl($block$)
            **for** $subBlock$ in $subBlocks$ in $reverse$ order **do**
               $deque.push\_front(subBlock)$
            **end for**
         **end if**
      **end while**
      **return** $blocks$
   **end procedure**

---

## 2.3  **LEGO-COMPILER**: THE FUNCTIONAL NEURAL COMPILER

We apply the LEGO translation method to the compilation domain and, guided by compilation-specific expert knowledge, design the **LEGO-Compiler**. This LLM-driven system accepts C programs as input and generates assembly code for x86, ARM, or RISC-V architectures. An overview of the LEGO-Compiler is depicted in Figure 2, primarily including the following Chain-of-Thoughts, where their detailed prompts can be found in Figure 11.

### 2.3.1  CONTROL FLOW ANNOTATION

The first annotation process addresses the positioning issues. Control flow in high-level languages consists of structures like if, while, for, and switch statements, which are linearized into branch-label constructs in assembly. In modern programming languages like C, most control statements (except goto) are encapsulated, meaning their generated labels remain within their scope. This property makes C programs composable at the control block level, which is formally proved in Theorem 5.

Control flow annotation is where **Part Split** is performed. Algorithm 1, which is also inspired by the composability proof algorithm in Figure 2, describes how to use LLM to split program into reasonably sized blocks. For small programs or control statements with low nesting levels, splitting may be unnecessary. For deeply nested control statements, further splitting may be required. This process maintains composability and encapsulation, ensuring correctness when rebuilding the full translation. In extreme cases, the program can be divided into basic blocks (Definition 3) or even sequential statements (Definition 2).

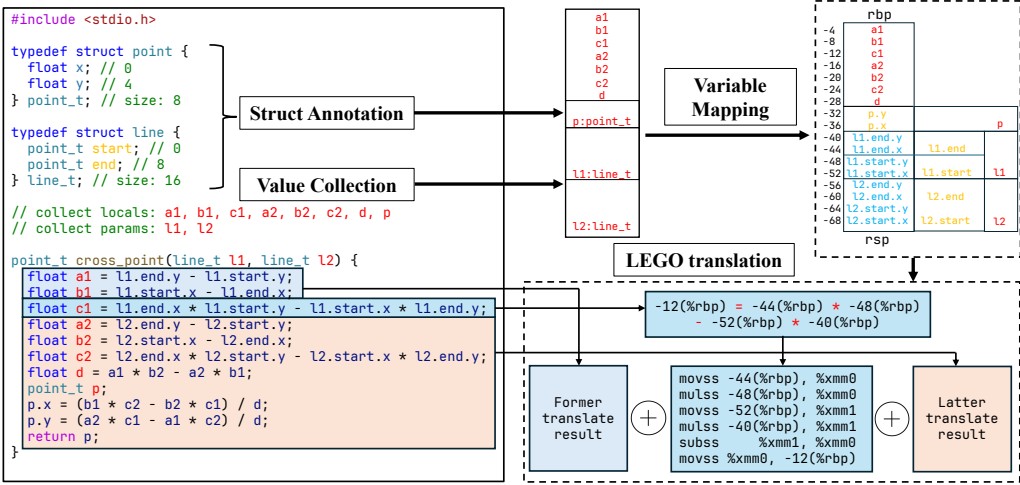

Figure 2: Workflow of **LEGO-Compiler**'s annotation-based Chain-of-Thoughts, **Struct Annotation** reasons type information from basic types, **Value Collection** finds all instances of each type, **Variable Mapping** explicitly bind variables from src language to dst language, then Part Split will split the program into composable parts, then **LEGO translate** them.

### 2.3.2 STRUCT ANNOTATION

The C language type system includes numerous basic types based on integer or float, their corresponding pointers, and compound types such as struct, union, and array, composed of basic types. Translating basic types and their instructions is relatively simple, as LLMs have learned this knowledge through extensive pre-training. However, for compound types, like structs (and similarly unions), the challenge arises from the infinite possible combinations of basic types.

To address this, we adopted a Chain-of-Thought approach. Instead of directly memorizing the mappings of variables from source-assembly language pairs in pretraining stage, we prompt the LLM to perform a separate thought process to reason about the memory layout of compound types based on structure, which includes size, offset, and alignment. Since compound types are ultimately composed of basic types, and LLMs understand basic type memory layouts, this pass can effectively infer the memory layout of compound types, like struct and array.

The Struct Annotation result is also verifiable using front-end tools like IntelliSense (Microsoft Corporation, 2024) or Clangd (LLVM Project, 2024b). After Struct Annotation, we obtain the symbol attributes for each type in the symbol table (LLM's context).

### 2.3.3 VARIABLE ANNOTATION

After type attribute inference, we need to determine where each type appears in the program and which variable identifier represents it. Our designed prompt guides the LLM to infer all declarations and arrange variable stack allocations according to their declaration order

Using x86 assembly as an example, global variables are stored in the data segment and indexed by same-name labels, providing clear binding relationships. For local variables, our method involves treating them as stored on the stack, assigning each a specific offset relative to a base address. By iterating through variable definitions, we can update these offsets relative to the base address, achieving effective stack allocation in most cases.

For compound type variables like structs, following the System V ABI (2018) for x86 assembly, access to sub-elements is achieved by adding offsets. This process is generally accurate with LLMs, relying primarily on precise binary integer arithmetic operations. After the Variable Annotation pass, we obtain a correspondence between variables in C and assembly languages, allowing for simple substitution during compilation, as illustrated in Figure 2.

An additional challenge is ensuring variable name uniqueness in the source program. We address this through a renaming pass at the source program level, eliminating name conflicts and ensuring variable uniqueness. This process is also verifiable through behavioral validation of the program before and after renaming.

### 2.3.4 SELF-CORRECTION THROUGH ERROR FEEDBACK

To address potential errors in the LLM-generated code, we implement a comprehensive self-correction mechanism. This system classifies errors into three categories: assembly semantic errors (detected by the assembler), runtime errors (identified through execution and caught by debuggers like gdb), and behavioral errors (discovered through result comparison). The error information is collected and fed back to the LLM for self-correction. Assembly semantic errors are typically straightforward to fix, while runtime errors, often caused by null pointer dereferences, are addressed by tracing instructions step-by-step to pinpoint the problematic area. Behavioral errors, being the most complex, may require multiple iterations to resolve. This iterative feedback and correction process significantly enhances the robustness and accuracy of the LLM-based compilation system, which is evaluated in the following section.

In general, LEGO-Compiler uses Chain-of-Thoughts in neural compilation task by either explicitly annotating the source code or generating intermediate results. These annotations and results are stored in the LLM's context, allowing the model to integrate them effectively. Through in-context learning, LEGO-Compiler is able to perform neural compilation tasks step by step, with each step being a simpler subtask that the LLM can handle. For more details, see subsection D.1.

## 3 EXPERIMENTS

To evaluate the effectiveness of our LEGO-Compiler approach, we have conducted a comprehensive set of experiments using three distinct datasets: ExeBench, CoreMark, and LongFunction. Each dataset serves a specific purpose in assessing different aspects of our neural compilation method.

### 3.1 EXPERIMENTAL SETUP

Major parameters we have tested are listed below, All settings use a one-shot prompt to help align the format. We evaluate the **Pass@k** correctness through IO unittests, altering the following settings, note that not all combinations of experimental settings are tested due to resource constraints.

- **models**: Advanced LLMs: GPT-4o (OpenAI, 2024), Claude-3.5-sonnet (Anthropic, 2023), Deepseek-coder (Guo et al., 2024) and Mini LLMs: GPT-4o-mini, Claude-3-haiku, and Codestral-22b (AI, 2024). We also test the newest o1-preview model (OpenAI, 2024) for limited evaluation.
- **method ablation**: Direct(baseline), annotation, annotation + fixing, annotation + fixing + LEGO translation, annotation + fixing + LEGO translation + pass@k(LEGO-Compiler)
- **T**emperature: 0.0-1.0, with 0.2 step increments
- **k**: 1, 5
- **fix rounds**: 0, 1, 3
- **architecture**: **x86**_64, **arm**-v8a, **riscv**64, majorly on **x86**

### 3.2 EXEBENCH EVALUATION

ExeBench (Armengol-Estapé et al., 2022) is a large-scale dataset of executable C programs, each equipped with a comprehensive unittest system. We use its Real-Executable subset, initially containing 40k samples. After data cleaning and removing samples that couldn't be compiled by the oracle compiler, our final test set consists of 23k samples. We utilize a test set of 500 cases randomly chosen from the full dataset for comprehensive evaluation due to resource and time constraints. To ensure the representativeness of this test set, we conduct additional evaluations on a larger subset of 5,000 cases using DeepseekCoder, one of our evaluating LLMs, whose results are consistent with those obtained from the smaller subset. We evaluate ExeBench through the following methodology:

1. Translate the C program to assembly using the LLM (to generate hypothesis), where we have three methods: direct, annotation-based CoT and LEGO translation.

2. Assemble and link the hypothesis assembly to create an executable.

3. Run the executable through 10 different IO test cases provided by ExeBench.

4. Consider the translation *successful* if it passes all test cases.

5. If a translation fails, apply self-fixing with the collected error feedback, will try **fix rounds**.

6. If still unsuccessful, proceed to the next iteration in **Pass@k**, until **k** is reached.

7. Consider the translation *failed* if it doesn't pass after all configured attempts.

The overall results on ExeBench are presented in Table 1, and case-difficulty ablation results in Figure 3, where we have tested 3 advanced LLMs and 3 mini LLMs respectively with the following experimental settings. Additionally, we compare the results with Zhang et al. (2024), the current state-of-the-art in neural compilation task, where their finetuned model is orthogonal to our work:

- Baseline: Direct code translation with Pass@1 and greedy-decoding. This represents the basic neural compilation capability of LLMs based on their default pretraining results. As we can see, models vary in a large margin, advanced models outperform mini models, and Claude-series outperform GPT series, where we found GPT series are facing simple syntax failures. Additionally, DeepseekCoder and Codestral performs well in the baseline setting.

- +Pass@k: Altering the **T**emperature to 0.6 and **k** to 5. This improves greatly for trivial code syntax errors as it allows LLMs trying different styles in the assembly, however, pass@k by allowing sampling on sub-optimal choices during decoding stage can only mitigate, but not solve the inefficient pretraining of LLMs. We see large improvement for all models, where most relatively simple cases are generated correctly during this stage, however, for harder cases, and those with pretraining biases(causing the errors), Pass@k is not helpful.

- +Feedback: Enabling the feedback self-fixing method, which enables LLMs to self-correcting its output from assembler feedback, runtime feedback and behavioral feedback. This significantly improves those cases with pretraining biases, because by explicitly providing error feedback, LLMs will reflect on their generation and focus on solving the errors. In comparison, assembler feedback is the most efficient feedback message, because it directly points out the errors; runtime feedback is helpful as well, though LLMs need to additionally reason its actual error occurrence from the message; behavioral message, since it lacks clear information about where is wrong in the hypothesis, although it's somehow helpful, LLMs' guesses on which part is problematic are usually wrong. In general, by enabling the feedback-driven LLM self-correction method, all models get significant improvement on their Pass@5 accuracy. Typically, the advanced LLMs majorly solve the simple and medium code, while the mini LLMs will still face some problems in simple cases.

- +CoT: Further enabling the annotation-based CoT methods described in subsection 2.3. This helps LLMs to reason the compilation process instead of direct generation, although it requires more tokens to be consumed as it generates intermediate text and reasoning steps. As a result, a large part of hard code is successfully generated, even for mini LLMs. The possible explanation is, these mini LLMs are not sufficiently pretrained on neural compilation datasets, but are sufficient for reasoning the logic of a guided compilation process, which is given by the annotation methods. Empirical results show all models pass at least 92.2% of the testset of ExeBench, which already looks good. Additionally, the 3 advanced LLMs reach around 99% IO Accuracy, showing that except for extreme hard cases, LLMs are sufficient to translate it well with guidelines of the annotation-based Chain-of-Thoughts.

- LEGO-Compiler(all): Coming so far, the remaining failed cases are all difficult in at least one of their features, detailed explanations of these difficulties are in subsection D.2. By translating a managed small part of code at a time and combining these results together to form a full neural compilation, LEGO-Compiler significantly solves the difficulty due to long code length and complicated control flows, where all advanced LLMs achieve over 99% accuracy in the ExeBench testset. Since the accuracy is very impressive, we further filter the hardest 10% subset of ExeBench based on the number of basic blocks and

Table 1: ExeBench experimental results

| Model | Baseline | +Pass@k | +Feedback | +CoT | LEGO-Compiler |
|---|---|---|---|---|---|
| GPT-4o | 76.8% | 93.4% | 97.8% | 99.2% | 99.8% |
| Claude-3.5-Sonnet | **92.6%** | **97.8%** | **98.6%** | **99.4%** | **100.0%**[1] |
| DeepseekCoder[2] | 82.48% | 87.96% | 93.76% | 97.36% | 99.24% |
| GPT-4o-mini | 58.8% | 74.8% | 86.0% | 92.2% | - |
| Claude-3-Haiku | 69.4% | **81.8%** | **90.0%** | **95.8%** | - |
| Codestral | **71.8%** | 79.4% | 88.6% | - | - |
| Finetuned LLM Compiler[3] | 91.72% | - | - | - | - |

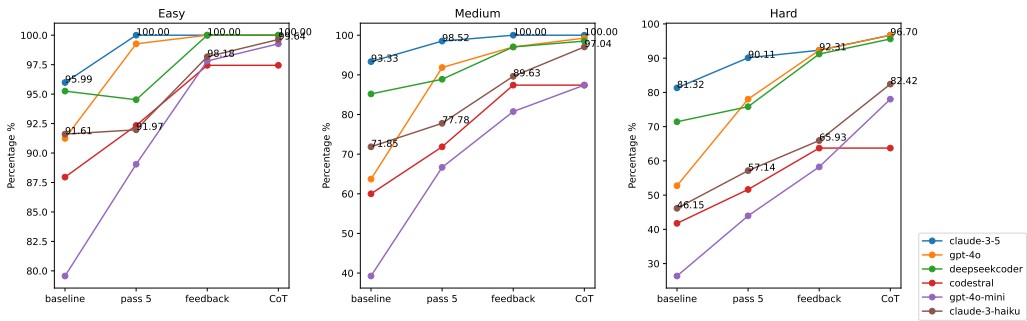

Figure 3: Ablation study: easy, medium and hard subsets ablations on ExeBench.

instructions within these blocks using the LLVM toolchain (LLVM Project, 2024a) to characterize the difficulty in ExeBench. The characterization of ExeBench and its hard subset is illustrated in Figure 6 in the appendix. Additionally, the experimental results on this hard subset, demonstrating the effectiveness of our methods, can be viewed in Table 2 also located in the appendix.

To sum up, the empirical results of our final LEGO-Compiler are suggesting the success of using LLMs for neural compilation, where the advanced LLMs solve almost all difficult cases in ExeBench(>99%), and the mini LLMs can also have over 95% accuracy.

## 3.3 CoreMark Evaluation, a case study

Previous evaluation has given promising results on neural compilation, with all methods applied, LLMs are achieving over 99% accuracy. However, considering ExeBench is a function-level compilation dataset that contains code with limited complexity, which we characterize in Figure 6, it is natural for us to think about applying our LEGO-Compiler for real-world codebases, where we choose CoreMark (Gal-On & Levy, 2012) as a case study, showing how complicated code that LLMs are capable of handling now.

CoreMark is a widely used benchmark for embedded devices, written entirely in C. It evaluates computer performance through state machine operations, linked list manipulations, and matrix computations. CoreMark consists of 40 functions, representing a complex, industry-grade codebase.

As depicted in Figure 4, the main function is one of the most complicated code in CoreMark, which contains a lot of complicated features of a C program. From another perspective, we can assert that if

---

[1] 100% test accuracy suggests LEGO-Compiler's state-of-the-art potential, but it doesn't ensure perfection in all scenarios, which suggests us to study harder cases as well.

[2] We evaluate the rest of models with a 500 subset of ExeBench, while we perform a larger scale 5000 subset of ExeBench with DeepseekCoder model, showcasing the consistency.

[3] We use the ExeBench results in the paper of Zhang et al. (2024) for comparison.

**"main" function in CoreMark**

```c
#include <stdint.h>
#include <stdio.h>
#include <stdlib.h>
#include <time.h>
/* custom structs */
typedef unsigned short u16;
typedef unsigned char u8;
typedef struct CORE_PORTABLE_S {
    u8 portable_id;
} core_portable;
typedef struct list_data_s {
    short data16;
    short idx;
} list_data;
typedef struct list_head_s {
    struct list_head_s *next;
    struct list_data_s *info;
} list_head;
typedef struct MAT_PARAMS_S {
    int N;
    short *A;
    short *B;
    int *C;
} mat_params;
typedef struct RESULTS_S {offset
    short seed1;              0 \
    short seed2;              2 \
    short seed3;              4 \
    void *memblock[4];        8  \
    unsigned size;            40  \
    unsigned iterations;      44   \
    unsigned execs;           48    \
    struct list_head_s *list;56     \
    mat_params mat;           64      \
    u16 crc;                  96
    u16 crclist;              98
    u16 crcmatrix;            100   /
    u16 crcstate;             102  /
    short err;                104 /
    core_portable port;       106 /
} core_results;       total:112 /
/* function declarations */
u16 crc16(short newval, u16 crc);
u16 crcu16(u16 newval, u16 crc);
void portable_init(core_portable *p);
void portable_fini(core_portable *p);
void *portable_malloc(size_t size);
void portable_free(void *p);
u8 check_data_types();
void *iterate(void *pres);
void start_time(void);
void stop_time(void);
clock_t get_time(void);
double time_in_secs(clock_t ticks);
list_head *core_list_init(unsigned blksize, list_head *memblock,
short seed);
void core_init_state(unsigned size, short seed, u8 *p);
unsigned core_init_matrix(unsigned blksize, void *memblk, int
seed,
mat_params *p);
/* global and static variables */
struct timespec start_time_val, stop_time_val;
static u16 list_known_crc[] = {(u16)0xd4b0, (u16)0x3340,
(u16)0x6a79,(u16)0xe714, (u16)0x3c1};
static u16 matrix_known_crc[] = {(u16)0xbe52, (u16)0x1199,
(u16)0x5608,(u16)0x1fd7, (u16)0x0747};
static u16 state_known_crc[] = {(u16)0x5e47, (u16)0x39bf,
(u16)0xe5a4,(u16)0x8e3a, (u16)0x8d84};
/* main function */
int main() {
    /* Part1:Prologue */
    /* Part2:Variable init */
    u16 i, j = 0, num_algorithms = 3;
    short known_id = -1, total_errors = 0;
    u16 seedcrc = 0;
    clock_t total_time;
    core_results results[1];
    portable_init(&(results[0].port));
    results[0].seed1 = 0;
    results[0].seed2 = 0;
    results[0].seed3 = 0x66;
    results[0].iterations = 0;
    results[0].execs = 7;
    results[0].size = 2000;
    results[0].memblock[0] = portable_malloc(results[i].size);
    results[0].err = 0;
    results[0].size = results[0].size / num_algorithms;
    /* Part3 */
    for (i = 0; i < 3; i++) {
        unsigned ctx;
        for (ctx = 0; ctx < 1; ctx++)
            results[ctx].memblock[i + 1] =
            (char *)(results[ctx].memblock[0]) + results[0].size * j;
        j++;
    }
    /* Part4 */
    results[0].list =
    core_list_init(results[0].size, results[0].memblock[1],
results[0].seed1);
    core_init_matrix(results[0].size, results[0].memblock[2],
(int)results[0].seed1 | (((int)results[0].seed2) << 16),
&(results[0].mat));
    core_init_state(results[0].size, results[0].seed1,
results[0].memblock[3]);
```

**Variable Mapping by LLM**

rbp

| | |
|---|---|
| i | -8 |
| j | -12 |
| n_algo | -16 |
| known_id | -20 |
| tot_errs | -24 |
| seedcrc | -28 |
| tot_time | -40 |
| port | -152+106 |
| err | |
| crcstate | |
| crcmatrix | ... |
| crclist | |
| crc | |
| mat | -152+64 |
| list | -152+56 |
| execs | |
| iterations | |
| size | |
| memblock | ... |
| seed3 | -152+4 |
| seed2 | -152+2 |
| seed1 results | -152+0 |
| ctx | -156 |
| sec_pass | -164 |
| divisor | -168 |

rsp

```c
/* Part 5 */
if (results[0].iterations == 0) {
    double secs_passed = 0;
    unsigned divisor;
    results[0].iterations = 1;
    while (secs_passed < (double)1) {
        results[0].iterations *= 10;
        start_time();
        iterate(&results[0]);
        stop_time();
        secs_passed = time_in_secs(get_time());
    }
    divisor = (unsigned)secs_passed;
    if (divisor == 0)
        divisor = 1;
    results[0].iterations *= 1 + 10 / divisor;
}
/* Part 6 */
start_time();
iterate(&results[0]);
stop_time();
total_time = get_time();
seedcrc = crc16(results[0].seed1, seedcrc);
seedcrc = crc16(results[0].seed2, seedcrc);
seedcrc = crc16(results[0].seed3, seedcrc);
seedcrc = crc16(results[0].size, seedcrc);
/* Part 7 */
switch (seedcrc) {
case 0xe9f5:
    known_id = 3;
    printf("2K performance run parameters for coremark.\n");
    break;
default:
    total_errors = -1;
    break;
}
/* Part 8 */
if (known_id >= 0) {
    results[i].err = 0;
    if ((results[i].execs & 1) &&
(results[i].crclist != list_known_crc[known_id])) {
        printf("[%u]ERROR! List crc 0x%04x - should be 0x%04x\n", i,
results[i].crclist, list_known_crc[known_id]);
        results[i].err++;
    }
    if ((results[i].execs & 2) &&
(results[i].crcmatrix != matrix_known_crc[known_id])) {
        printf("[%u]ERROR! matrix crc 0x%04x - should be 0x%04x\n",
i, results[i].crcmatrix, matrix_known_crc[known_id]);
        results[i].err++;
    }
    if ((results[i].execs & 4) &&
(results[i].crcstate != state_known_crc[known_id])) {
        printf("[%u]ERROR! state crc 0x%04x - should be 0x%04x\n",
i, results[i].crcstate, state_known_crc[known_id]);
        results[i].err++;
    }
    total_errors += results[i].err;
}
/* Part 9 */
total_errors += check_data_types();
printf("CoreMark Size : %lu\n", (long unsigned)results[0].size);
printf("Total ticks : %lu\n", (long unsigned)total_time);
printf("Total time (secs): %f\n", time_in_secs(total_time));
if (time_in_secs(total_time) > 0)
    printf("Iterations/Sec : %f\n",
1 * results[0].iterations / time_in_secs(total_time));
if (time_in_secs(total_time) < 10) {
    printf("ERROR! Must execute for at least 10 secs for a valid
result!\n");
    total_errors++;
}
printf("Iterations : %lu\n", (long
unsigned)results[0].iterations);
printf("Compiler version : AICC 1.0\n");
printf("seedcrc : 0x%04x\n", seedcrc);
/* Part 10 */
if (results[0].execs & 1)
    printf("[%d]crclist : 0x%04x\n", i, results[i].crclist);
if (results[0].execs & 2)
    printf("[%d]crcmatrix : 0x%04x\n", i, results[i].crcmatrix);
if (results[0].execs & 4)
    printf("[%d]crcstate : 0x%04x\n", i, results[i].crcstate);
    printf("[%d]crcfinal : 0x%04x\n", i, results[i].crc);
if (total_errors == 0) {
    printf("Correct operation validated. See README.md for run
and reporting rules.\n");
    if (known_id == 3)
        printf("Function Level CoreMark 1.0 : %f by AICC 1.0",
results[0].iterations / time_in_secs(total_time));
        printf(" / Heap");
        printf("\n");
}
if (total_errors > 0)
    printf("Errors detected\n");
if (total_errors < 0)
    printf("Cannot validate operation for these seed values,
please compare with results on a known platform.\n");
portable_free(results[0].memblock[0]);
portable_fini(&(results[0].port));

return 0;
/* Part 11: Epilogue */
}
```

Figure 4: The CoreMark main function, one of the most difficult code we evaluated. In this figure, all CoTs are illustrated in the code annotations in color, as well as the variable mapping process.

CoreMark's main function can be neural compiled, simpler code, can be neural-compiled correctly in high possibility.

In general, Claude-3.5-sonnet compiles all 40 out of 40 functions correctly, both GPT-4o and DeepseekCoder achieves 38, where they fail to generate the **main** function and another complicated **core_bench_state** function. The reason for their failure is not on the complicated code control structures, but on the translation of certain instructions, which can be improved with more compiler knowledge taught to them. If taught with such knowledge (manually prompted), all three LLMs can successfully compile the whole CoreMark, achieving functionality just as oracle compiler does.

### 3.4 LONGFUNCTION EVALUATION

Since the LEGO translation method significantly scales up the capability of long code translation, we design LongFunction, a synthesized dataset for testing very long code translations, particularly for evaluating the effectiveness of our LEGO translation method. The dataset is made up of 50 synthesized programs, ranging from 317 to 238737 tokens in length, each program is self-contained and can be compiled and run independently. We evaluate the neural compilation task on LongFunction dataset for all x86_64, arm-v8a and riscv64 architectures, and the neural code translation task by migrating C to Python/C++/Rust.

Examples of LongFunction and the evaluation details can be found in subsection D.3 due to page limits. In conclusion, our proposed LEGO translation method breaks the complexity of long code, boosting their capability of handling long code, where all LLMs we tested passed the whole Long-Function dataset. In comparison, the current best model, o1-preview, can only maximally translate a 5772 token sized case using direct translation method.

The results of LongFunction dataset evaluation give us a strong insight: Code, or programming languages, unlike the natural languages, no matter how long they are, their complexity can be divide-and-conquered into two levels: The first is on the control flow level, which combines each block of code logic together to form a long and complicated code, where both current compilers and our LEGO-Compiler methods can iteratively split the code into smaller and smaller, managable code snippets to overcome this complexity. The second is on the handling of each code snippet, where the size is not large, and the major difficulties are on how to correctly translate each statement of it correctly, where we also find out that for these advanced LLMs, they perform strongly on this level. Since the basic operations are also limited in programming languages, within proper size, the code snippet level translation or compilation can be gradually improved to near 100% with the advancement of more powerful pretrained LLMs.

## 4 CONCLUSION

We have presented LEGO-Compiler, a novel approach to neural compilation that leverages Large Language Models (LLMs) to translate high-level programming languages into assembly code. Our LEGO translation method breaks down large programs into manageable, self-contained blocks through the composable nature of code, significantly extending the scalability of neural code translation. By incorporating a series of Chain-of-Thought stages guided by classical compiler design and self-correction mechanisms, LEGO-Compiler effectively addresses key challenges in compilation tasks, achieving significant improvements in accuracy and scalability. Our experimental results demonstrate the effectiveness of LEGO-Compiler, as it achieves over 99% accuracy on the ExeBench dataset and fully compiles the industrial-grade CoreMark benchmark correctly. We also introduce LongFunction, a new dataset designed to evaluate the translation and compilation of lengthy code, demonstrating the effectiveness of the proposed LEGO translation method.

These findings provide important insights into the capabilities and limitations of LLMs in neural compilation tasks. While our current implementation incurs higher computational costs compared to traditional compilers, it offers unique advantages in adaptability and potential for rapid integration of new instruction sets or language features. As LLM capabilities continue to improve, approaches like LEGO-Compiler are poised to play an increasingly important role in the future of software development and compilation, complementing and enhancing traditional compiler technologies.

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

# A RELATED WORK

## A.1 CODE TRANSLATION

**Code Translation** has evolved from traditional statistical methods (Nguyen et al., 2015b) to neural-based approaches that capture programming language structures (Chen et al., 2018). Current neural code translation researches can be majorly categorized to two types: learning-based transpilers (Rozière et al., 2020; Roziere et al., 2021; Wen et al., 2022) and pre-trained language models (Feng et al., 2020; Wang et al., 2021; Lu et al., 2021; Rozière et al., 2022; OpenAI et al., 2023; Anthropic, 2023). The former majorly studies the scarcity of parallel corpora (Xie et al., 2023) and develops unsupervised learning methods to overcome it. The latter using Large Language Models' vast pretrained knowledge, can also perform code translations well without training (Yang et al., 2024; Liu et al., 2024a).

**Analysis of neural code translation** is equally crucial. Studies have examined common pitfalls in language models for code intelligence (She et al., 2023; Jiao et al., 2023), investigated the reliability and explainability of these models in automated program generation (Liu et al., 2024b), and the attention paid by LLM during code generation that differs from human (Kou et al., 2024).

As for **compilation related translations**, Armengol-Estapé & O'Boyle (2021) first gives a try on neural compilation. Guo & Moses (2022) further studies on C-to-LLVM IR translation. However, they only perform limited investigations on the methods, and their results are still preliminary. There are also works on the reverse decompilation process (Fu et al., 2019; Cao et al., 2022; Armengol-Estapé et al., 2023) and works on code optimizations (Cummins et al., 2023; 2024). The most related work is Zhang et al. (2024), which achieves state-of-the-art 91% Pass@1 accuracy compared to non-LLM methods on the C-x86 task using a finetuned CodeLlama model, which our work surpasses. Besides, their approach relies on compiler-generated bilingual corpora, while our methods can effectively eliminate such dependency by reasoning the steps of how a compiler works.

Finally, the **breakdown of neural code translation** is also less studied, Nguyen et al. (2015a) first breaks the translation of Java-C# into syntaxemes level to lower the translation difficulty in SMTs. Our work uses similar divide-and-conquer methodology to breakdown a large long code into manageable control block parts, then LLMs can translate these parts separately with the aid of necessary context and combine their results into a large, complete and coherent translation.

## A.2 OTHER RELATED WORK

**LLM self-repair.** Recent research has focused extensively on enhancing LLMs' self-correction capabilities. Several studies closely related to our work deserve mention. A comprehensive survey by Pan et al. (2024) thoroughly examined methods for leveraging feedback to autonomously improve LLM outputs. Wang et al. (2022b) first uses compiler feedback for better code generation, and Dou et al. (2024) establishes the syntax-runtime-functional bug type taxonomy and builds corresponding self-repair pipelines for code. Our work is their natural extensions to neural compilation scenario. While Olausson et al. (2024) investigated the limitations of self-repair mechanisms in code generation, our findings diverge significantly. Contrary to their conclusions, we discovered that self-repair serves as a highly effective solution in the neural compilation process, particularly when incorporating syntax feedback and runtime feedback.

**In-context learning and Chain-of-Thoughts.** LLMs are able to in-context learn via inference alone by providing few shots of demonstration then predicting on new inputs (Min et al., 2022; Dong et al., 2024). Thus customized Chain-of-Thoughts (Wei et al., 2022; Chu et al., 2024) can guide LLMs to perform complicated reasoning (Wang et al., 2022a; Song et al., 2024), which is the cornerstone of our work. More specifically, Levy et al. (2024) reveals the degradation of LLMs' performance for long context, and validate the effectiveness on using Chain-of-Thoughts to mitigate. We found similar results in code translation/compilation tasks. However, our proposed **LEGO translations** method can significantly mitigate such degradation as it turns a long context direct translation into multiple composable, shorter ones that LLMs can handle.

**Generation Scalability and Long Context Learning.** Except for code translation, many LLM-based method will fall into scalability problems since larger inputs are not well trained like the smaller ones. So methods to extend LLMs scalability remain an interesting study. For example,

in order to coherently generate long passages of text, Tan et al. (2021) proposes a multi-staged keyword-first progressive method to improve it significantly, where our work shares a similar insight. Li et al. (2024) introduces a self-route method to dynamically choose the usage of RAG or fully in-context, balancing the cost and performance in long-context scenario, which inspires us to use an analyze-first, then-CoT approach.

Needle-in-the-haystack experiment (Kuratov et al., 2024) is a well-known test for testing LLMs' capability for long context, however, it only requires simple reasoning on the needle part, where the test is not complicated enough. There are more works evaluating the long-context learning capabilities of LLMs. Vodrahalli et al. (2024) examines LLMs with their proposed Latent Structure Queries evaluation, which aims to chisel away irrelevant information in the context, revealing a latent structure in the context, which provide a stronger signal of long-context language model capabilities. Prompt compression is another useful method to improve the long-context inference capabilities (Jha et al., 2024), which is widely used for retrieval-augmented generation(RAG) systems by compressing the long contexts. Our work and its broader area: neural compilation/translation in large codebase, could serve as another useful real-world application for long-context inference.

# B    COMPOSABILITY OF C-LIKE LANGUAGE CONSTRUCTS

## B.1    DEFINITIONS AND LANGUAGE STRUCTURE

We define a simplified C-like language structure using the following EBNF-inspired grammar:

```
block: '{' (blockItem)* '}';
blockItem: decl | stmt;
stmt:
    lVal '=' exp ';'                        # assignStmt
    | exp ';'                               # exprStmt
    | 'goto' label ';'                      # gotoStmt
    | ';'                                   # blankStmt
    | block                                 # blockStmt
    | IF '(' exp ')' stmt (ELSE stmt)?      # ifStmt
    | WHILE '(' exp ')' stmt                # whileStmt
    | FOR '(' stmt exp ';' stmt ')' stmt    # forStmt
    | SWITCH '(' stmt ')' stmt              # switchStmt
    | BREAK ';'                             # breakStmt
    | CONTINUE ';'                          # continueStmt
    | RETURN (exp)? ';'                     # returnStmt;
```

We derived from the grammar that describes C-like language to form the following definitions. Also for simplicity purposes, we omit the slight differences between **decl**, **stmt** and **exp**.

**Definition 2** (Basic Statement). *A basic statement is a statement that does not contain any other statements within its structure. This includes assignStmt, exprStmt, gotoStmt, blankStmt, breakStmt, continueStmt, and returnStmt. We first exclude gotoStmt for the main proof for simplicity.*

**Definition 3** (Basic Block). *A basic block is a sequence of consecutive basic statements as defined in Definition 2, in which flow of control enters at the beginning and leaves at the end without halt or possibility of branching except at the end.*

**Definition 4** (Control Block). *A control block is a code snippet that reflects a complete control structure, such as for(;;){}, if(){}[else{}], while(){}, do{}while(), or switch(){case:...}. Each subpart of a control block can be other control blocks or basic blocks as defined in Definition 3.*

**Definition 5.** *A basic control block is an innermost control block (Definition 4) where each of its subparts contains only basic blocks as defined in Definition 3.*

**Definition 6** (Compound Control Block). *A compound control block is a control block (Definition 4) that contains at least one subpart that is not a basic block (Definition 3), but rather another control block as defined in Definition 4.*

**Definition 7** (Translation Function and Valid Translations). *Let $\mathcal{T}$ be the set of all valid translation functions from $SRC$ to $DST$, where $SRC$ is the source language (our C-like language) and $DST$ is the destination language (e.g., x86 assembly).*

*Formally, $\mathcal{T} = \{T \mid T : SRC \rightarrow DST\}$ such that for any $T \in \mathcal{T}$ and any $stmt \in SRC$:*

*1. $T(stmt) \in DST$ 2. $T(stmt)$ preserves the semantics of $stmt$*

*A translation function $T \in \mathcal{T}$ maps each construct in the source language to one or more constructs in the destination language while preserving the program's behavior.*

**Definition 8** (Translation Composability). *Let $(SRC, \circ)$ be the source language with concatenation operation $\circ$, and $(DST, \cdot)$ be the destination language with concatenation operation $\cdot$. Let $\mathcal{T}$ be the set of valid translation functions as defined in Definition 7.*

*Translation composability holds if and only if:*

$$\exists T \in \mathcal{T} : \forall P_1, P_2 \in SRC, T(P_1 \circ P_2) \equiv T(P_1) \cdot T(P_2)$$

*Where:*

- *$T : SRC \rightarrow DST$ is a translation function*

- *$\equiv$ denotes semantic equivalence, preserving both control flow and data flow*

- *$\circ : SRC \times SRC \rightarrow SRC$ is the concatenation operation in the source language*

- *$\cdot : DST \times DST \rightarrow DST$ is the concatenation operation in the destination language*

B.2 COMPOSABILITY OF BASIC STATEMENTS

**Theorem 1** (Composability of Basic Statements). *For any two basic statements $stmt_1$ and $stmt_2$ in SRC, as defined in Definition 2, their translation is composable: $T(stmt_1 \circ stmt_2) \equiv T(stmt_1) \cdot T(stmt_2)$*

*Proof.* We prove this for all combinations of assignment statements and expression statements. The proof considers control flow preservation, data flow preservation, and independence of translation. Other basic statements (blank, return, etc.) trivially maintain composability as they do not affect control or data flow when composed with other basic statements. □

**Example B.1.** *This example illustrates the composability of basic statements as defined in Definition 2 and proved in Theorem 1.*

*Consider the following sequence of basic statements:*

```
a = b + 3;  // stmt_1
b = a - 1;  // stmt_2
```

*The translation of these statements might look like:*

```
T(stmt_1):
    mov eax, [b]
    add eax, 3
    mov [a], eax

T(stmt_2):
    mov eax, [a]
    sub eax, 1
    mov [b], eax
```

*These translations are composable because:*

*1. Control Flow: The order of execution is preserved (stmt_1 then stmt_2). 2. Data Flow: The value of 'a' computed in stmt_1 is correctly used in stmt_2. 3. Independence: The translation of stmt_2 does not depend on how stmt_1 was translated, only on its effect (the value of 'a').*

*Therefore, $T(stmt_1 \circ stmt_2) \equiv T(stmt_1) \cdot T(stmt_2)$, demonstrating composability.*

Example B.1 illustrates that even when statements have data dependencies, their translations remain composable as long as the order of operations is preserved. Similar proof of composability can be made for all stmts within a basic block (Definition 3).

### B.3 COMPOSABILITY OF BASIC CONTROL STRUCTURES

**Theorem 2** (Composability of Basic Control Structures). *Basic control structures (if-else, for, while, do-while, switch-case), where all their components are basic blocks as defined in Definition 3, are composable under the translation function $T$ as defined in Definition 7.*

*Proof.* We will prove this for each basic control structure:

1. For Loop:

Let $B_{init}$, $B_{cond}$, $B_{incr}$, and $B_{body}$ be the basic blocks for init, cond, incr, and body respectively.

Translation structure:

```
T(basic_for_loop):
    T(B_init)
loop_start:
    T(B_cond)
    jz loop_end
    T(B_body)
    T(B_incr)
    jmp loop_start
loop_end:
```

1. Control Flow Preservation: The structure of jump instructions preserves the original control flow. 2. Data Flow Preservation: The order of operations within and between blocks is maintained. 3. Composability: $T(basic\_for\_loop) \equiv T(B_{init}) \cdot T(B_{cond}) \cdot T(B_{body}) \cdot T(B_{incr})$, where $\cdot$ represents concatenation with appropriate jump instructions.

Therefore, the basic for loop is composable under $T$. Similar proofs can be constructed for other basic control structures. $\square$

2. If-Else Statement: Let $B_{cond}$, $B_{then}$, and $B_{else}$ be the basic blocks for condition, then-branch, and else-branch respectively.

Translation structure:

```
T(basic_if_else):
    T(B_cond)
    jz else_label
    T(B_then)
    jmp end_label
else_label:
    T(B_else)
end_label:
```

Control flow and data flow preservation follow similarly to the for loop case.

3. While Loop: Let $B_{cond}$ and $B_{body}$ be the basic blocks for condition and body respectively.

Translation structure:

```
T(basic_while):
loop_start:
    T(B_cond)
    jz loop_end
    T(B_body)
    jmp loop_start
```

```
loop_end:
```

4. Do-While Loop: Let $B_{body}$ and $B_{cond}$ be the basic blocks for body and condition respectively.

Translation structure:

```
T(basic_do_while):
loop_start:
    T(B_body)
    T(B_cond)
    jnz loop_start
```

5. Switch-Case Statement: Let $B_{expr}$ be the basic block for the switch expression, and $B_1, B_2, ..., B_n$ be the basic blocks for each case.

Translation structure:

```
T(basic_switch):
    T(B_expr)
    cmp result, case1_value
    je case1_label
    cmp result, case2_value
    je case2_label
    ...
    jmp default_label
case1_label:
    T(B_1)
    // No break implies fall-through
case2_label:
    T(B_2)
    ...
default_label:
    T(B_n)
end_switch:
```

For all these structures, control flow is preserved by the appropriate use of jump instructions, and data flow is maintained by the sequential execution of basic blocks. The translation of each structure is a composition of its basic block translations, proving composability.

**Theorem 3** (Composability of Break and Continue Statements). *Break and continue statements, which are basic statements as per Definition 2, are composable within their respective control structures when proper loop depth tracking is maintained.*

*Proof.* Let $loop\_depth$ be a counter maintained during translation to track nested loop levels.

1. Break Statement: Translation structure:

```
T(break):
    jmp loop_end_label_depth
```

Where $loop\_end\_label\_depth$ corresponds to the end of the current loop at depth $loop\_depth$.

2. Continue Statement: Translation structure:

```
T(continue):
    jmp loop_continue_label_depth
```

Where $loop\_continue\_label\_depth$ corresponds to the continuation point of the current loop at depth $loop\_depth$.

Control flow is preserved by jumping to the appropriate label based on the current loop depth. Data flow is trivially preserved as these statements do not modify data.

---

**Algorithm 2** Iterative Bottom-Up Composability Proof Algorithm

---

**procedure** PROVECOMPOSABILITY(Program $P$)
    $blocks \leftarrow$ DecomposeIntoOutermostControlBlocks($P$)              ▷ Initial decomposition
    $to\_process \leftarrow$ new Deque()
    **for** each $block$ in $blocks$ **do**
        $to\_process$.PushBack($block$)              ▷ Initialize processing queue
    **end for**
    **while** $to\_process$ is not empty **do**
        $current\_block \leftarrow to\_process$.PopFront()       ▷ Handle first unhandled block
        **if** IsBasicBlock($current\_block$) **then**
            **continue**                         ▷ Do nothing
        **else if** IsControlStructure($current\_block$) **then**
            $sub\_blocks \leftarrow$ SplitControlStructure($current\_block$)
            **for** each $sub\_block$ in $sub\_blocks$ in $reverse$ order **do**
                $to\_process$.PushFront($sub\_block$)     ▷ Handle sub-blocks in original order
            **end for**
        **else**
            **return** $P$ is not composable               ▷ Unrecognized structure
        **end if**
    **end while**
    **return** $P$ is composable
**end procedure**
**function** SPLITCONTROLSTRUCTURE(Block $b$)
    **if** $b$ is a For Loop **then**
        **return** SplitForLoop($b$)
    **else if** $b$ is an If-Else structure **then**
        **return** SplitIfElse($b$)
    **else**
        **return** SplitOtherControlStructure($b$)     ▷ Extensible for other structures
    **end if**
**end function**
**function** SPLITFORLOOP(ForLoop $f$)           ▷ Decompose for loop into constituent parts
    **return** [ $f.init$, $f$.ForBodyLabel, $f.cmp$, ConditionalJump($f$.ForEndLabel),
$f.body$, $f.incr$, UnconditionalJump($f$.ForBodyLabel), $f$.ForEndLabel ]
**end function**
**function** SPLITIFELSE(IfElse $i$)           ▷ Decompose if-else into constituent parts
    **return** [ $i.cmp$, ConditionalJump($i$.ElseLabel), $i.then\_body$,
UnconditionalJump($i$.EndIfLabel), $i$.ElseLabel, $i.else\_body$, $i$.EndIfLabel ]
**end function**

---

The composability of these statements within their containing loops is maintained because: a) They generate a single jump instruction that integrates with the loop's control flow. b) The loop depth tracking ensures the jump targets the correct loop level in nested structures.    □

### B.4 COMPOSABILITY OF COMPLEX STRUCTURES

**Definition 9** (Composable Control Block). *A composable control block is either:*

- *A basic block as defined in Definition 3, or*

- *A basic control structure as proved in Theorem 2, or*

- *A sequence of composable control blocks, or*

- *A control structure whose all subparts are composable control blocks.*

**Theorem 4** (Composability of Sequential Control Blocks). *A sequence of composable control blocks $CB_1, CB_2, ..., CB_n$ as defined in Definition 9 is composable under the translation function $T$.*

*Proof.* Let $CB_1, CB_2, ..., CB_n$ be composable control blocks. 1. By Definition 9, each $CB_i$ is composable. 2. Translation structure: $T(CB_1 \circ CB_2 \circ ... \circ CB_n) \equiv T(CB_1) \cdot T(CB_2) \cdot ... \cdot T(CB_n)$ where $\circ$ denotes sequential composition in SRC and $\cdot$ denotes concatenation in DST. 3. Control Flow Preservation: The sequential order of control blocks is maintained in the translation. 4. Data Flow Preservation: The order of operations between control blocks is preserved.

Therefore, the sequence of composable control blocks is itself a composable control block under $T$. □

**Theorem 5** (Composability of Arbitrary Programs). *Any program $P$ that can be decomposed into a sequence of control blocks as defined in Definition 4 is composable under the translation function $T$ if the Iterative Composability Proof algorithm (Figure 2) marks it as composable.*

*Proof.* The proof follows from the correctness of the Iterative Composability Proof algorithm:

1. The algorithm starts with basic blocks and basic control structures, which are proven composable by Theorem 1 and Theorem 2.

2. It iteratively builds up composability for larger structures:

   - Sequences of composable blocks are proved composable by Theorem 4.
   - Control structures with all composable subparts are marked composable.

3. The process continues until the entire program is marked composable or no further progress can be made.

4. If the entire program is marked composable, it means that $T(P)$ can be expressed as a composition of the translations of its composable parts, preserving both control flow and data flow as per Definition 8.

Therefore, if the algorithm returns that $P$ is composable, then $P$ is indeed composable under the translation function $T$. □

**Theorem 6** (Composability of Goto Statements). *Goto statements, which are basic statements as per Definition 2, are composable under the translation function $T$, but aribitrary goto statements can break the structured control flow assumed in the main proof.*

*Proof.* Let $l$ be a label and $goto\ l$ be a goto statement.

Translation structure:

```
T(goto l):
    jmp label_l

T(l:):
label_l:
```

The goto statement translates to an unconditional jump, preserving control flow. It doesn't directly affect data flow. Composability holds as $T(stmt_1 \circ goto\ l \circ stmt_2) \equiv T(stmt_1) \cdot T(goto\ l) \cdot T(stmt_2)$.

However, goto introduces complications:

- Non-local control flow can break the nested structure of control blocks.

- Programs with unrestricted goto usage are difficult to decompose into well-defined control blocks.

- It can lead to unstructured code, complicating reasoning about program behavior.

$\square$

While goto is provably composable, it's discouraged in modern programming for readability, maintainability, and optimization reasons. Our composability principle is most applicable and valuable in the context of structured programming paradigms.

### B.5 SCOPE AND LIMITATIONS OF THE PROOF

The proof of composability presented in this paper is based on a simplified model of C-like languages and unoptimized translation. It's important to note several key points about the scope and limitations of this proof:

1. **Simplification and Correctness:** The simplifications made in our language model and translation process do not compromise the validity of the proof. The core of our argument relies on the decomposition of programs into control blocks and the composability of these blocks. The internal structure of basic blocks, while important for actual compilation, does not affect the composability principle we've established.

2. **Unoptimized Translation:** Our proof assumes a straightforward, unoptimized translation process. This assumption is crucial for maintaining the direct correspondence between source code structures and their translations.

3. **Limitations for Complex Language Features:** The composability principle as proved here can be applied to C-like languages, but may not hold for more complex language features. For example:
   - Exception Handling: Languages with sophisticated exception handling mechanisms, such as Python, introduce complexities that can break composability. These mechanisms often require:
     - Guarded execution of code blocks.
     - Runtime type information (RTTI) for determining appropriate exception handlers.
     - Non-local control flow that can't be easily decomposed into our model of control blocks.
   - Coroutines and Generators: Features that allow for suspending and resuming execution mid-function can introduce state that is not easily captured in our model of control flow.
   - Reflection and Metaprogramming: Languages that allow for runtime modification of program structure or behavior can invalidate static composability assumptions.

   Although not applicable to some specific language features, it doesn't mean the composablity and its derived LEGO translation method is not applicable to the whole programming language, as long as these features are not used in the code, the composability will still stand and the LEGO translation will still work.

4. **Optimizations Across Basic Blocks:** Our proof assumes that the boundaries of control blocks are respected in the translation process. However, many real-world compiler optimizations operate across these boundaries. Examples include:
   - Loop unrolling
   - Function inlining
   - Global value numbering
   - Code motion optimizations

   Such optimizations can reorder, eliminate, or combine operations from different control blocks, potentially breaking the composability property as we've defined it.

5. **Applicability:** Despite these limitations, the composability principle proved here is valuable for:
   - The foundation of LEGO translation method, the proof reveals the composable nature of code in at least control block level, which is a major difference than natural languages.

- The proof process also guided Algorithm 1 in LEGO translation, as proving the composability and making use of the composability share similar algorithms.

In conclusion, while our proof provides a strong foundation for understanding composability in C-like languages with straightforward translation, it's important to recognize its boundaries. More complex language features may require extensions or modifications to this framework to maintain composability guarantees. And optimized code translation usually is not composable.

# C  DISCUSSIONS

## C.1  UNIVERSALITY OF LEGO TRANSLATION

The LEGO translation method, while initially developed for compilation tasks, demonstrates broader applicability based on fundamental properties of programming languages rather than being specific to compilation. The composability that LEGO translation leverages stems from the well-encapsulated control flow and locality principles inherent in modern programming languages (disregarding constructs like **goto** in C, more limitations are clearly described in Appendix E).

These characteristics are intrinsic to programming languages themselves and have guided modern compiler design. They enable the modular partitioning of large-scale programs in modern software development, allowing for incremental and even parallel compilation of code. We harness these properties and apply them to the context of neural compilation using Large Language Models (LLMs).

It's important to note that the applicability of LEGO translation extends beyond compilation. It is suitable for various tasks originating from programming languages, such as code translation between different languages. This method significantly enhances the scalability of machine translation tasks for code, providing a powerful tool for handling large and complex codebases.

## C.2  MANAGING HIGHLY COMPLEX EXPRESSIONS

One of the primary challenges in neural compilation arises when dealing with expressions or statements of high complexity. In such cases, LLMs struggle to accurately evaluate these expressions through next token prediction. To address this, we propose two solutions:

- External Tool Integration: We can utilize external parsing tools to generate tree structure information for complex expressions evaluation. This tree structure is then provided to the LLM, offering an explicit traversal order and guiding the evaluation process.

- Expression Decomposition: Without relying on external tools, we can design a new pass where the LLM identifies high-complexity expressions and rewrites them as a combination of lower-complexity expressions. This approach ensures that the entire program consists only of expressions within a proper LLM's evaluation capabilities.

## C.3  COMPUTATIONAL COST, EFFECTIVENESS, AND FUTURE PROSPECTS

While our neural compilation method is primarily a proof of concept, it does incur significantly higher computational costs compared to traditional compilation methods - approximately $10^6$ to $10^7$ times higher. However, this should be weighed against the substantial human resources required for traditional compiler development.

The key advantage of our approach lies in its potential for rapid adaptation to new instruction set extensions or frontend intrinsics. Through techniques like RAG (Retrieval-Augmented Generation) and in-context learning, our method can be extended to support new architectures or language features. This positions neural compilation as a valuable assistant in the compiler development process. A particularly promising application is in generating end-to-end unit tests for compiler adaptation to new instructions. This could significantly streamline the development and testing phases of compiler updates. Recent researches like Munley et al. (2024) have shown the ability to using LLMs to generate unittests during compiler validations.

**Input:**

**System Message:** you are a professional AI assistant in code, based on the user input C code, you are going to help me to generate the corresponding x86 assembly.
You will perform like a compiler with O0 optimization level, the architecture is x86_64. You should think carefully, follow the guidelines.
**Guide1:** Analyze the input C code, determine whether it contains the following program attributes, return the key name if you are certain.
**"numerical"**: If the code contains numerical values, like 1.0, 2e-5, 3.14f, etc, if the code only use integers, then don't include this feature.
**"hex_octal"**: If the code contains hex or octal values, like 0x3f, 077, etc.
**"funcall"**: If the code contains other function calls.
**"recursive"**: If the code function is recursive.
**"long"**: If the code is long and complex (more than 50 lines)
**"str"**: If the code manipulates strings or char arrays.
**"order"**: If the code contains complicated expressions (need many operations to evaluate), be aware of the order of operations.
**Guide2:** For these triggered features, do more thoughts.
**IO Format:** input code will be inside "```c" and "```"tags, please also make sure the generated x86 assembly be inside "```x86" and "```" tags.
**Minimal Example:**

```
#Input:
```c
#include <stdio.h>
int main() {
printf("Hello, World!\n");
return 0;
}
```
```

```
#Output:
```x86
    .text
    .globl main
    .type main, @function
main:
.L_main_entry:
    endbr64
    pushq %rbp
    movq %rsp, %rbp
    leaq .LC0(%rip), %rdi
    call printf@PLT
    movl $0, %eax
    popq %rbp
    ret
.L_print_hello:
    .string "Hello, World!"
```
```

```
#Input:
```c
double interp_weno7(double phim3, double phim2, double phim1, double phi,
double phip1, double phip2, double phip3){
const double p0 = (-1.0/4.0)*phim3 + (13.0/12.0) * phim2 + (-23.0/12.0) *
phim1 + (25.0/12.0)*phi;
const double p1 = (1.0/12.0)*phim2 + (-5.0/12.0)*phim1 + (13.0/12.0)*phi
+ (1.0/4.0)*phip1;
const double p2 = (-1.0/12.0)*phim1 + (7.0/12.0)*phi + (7.0/12.0)*phip1 +
(-1.0/12.0)*phip2;
const double p3 = (1.0/4.0)*phi + (13.0/12.0)*phip1 + (-5.0/12.0)*phip2 +
(1.0/12.0)*phip3;
const double beta0 = (phim3*(547.0*phim3 - 3882.0*phim2 + 4642.0*phim1 -
1854.0*phi)
+ phim2*(7043.0*phim2 - 17246.0*phim1 + 7042.0*phi)
+ phim1*(11003.0*phim1 - 9402.0*phi)
+ 2107.0*phi*phi);
const double beta1 =(phim2*(267.0*phim2 - 1642.0*phim1 + 1602.0*phi -
494.0*phip1)
+ phim1*(2843.0*phim1 - 5966.0*phi + 1922.0*phip1)
+ phi*(3443.0*phi - 2522.0*phip1)
+ 547.0*phip1*phip1);
const double beta2 = (phim1*(547.0*phim1 - 2522.0*phi + 1922.0*phip1 -
494.0*phip2)
+ phi*(3443.0*phi -5966.0*phip1 + 1602.0*phip2)
+ phip1*(2843.0*phip1 - 1642.0*phip2)
+ 267.0*phip2* phip2);
const double beta3 = (phi*(2107.0*phi - 9402.0*phip1 + 7042.0*phip2 -
1854.0*phip3)
+ phip1*(11003.0*phip1 - 17246.0*phip2 + 4642.0*phip3)
+ phip2*(7043.0*phip2 - 3882.0*phip3)
+ 547.0*phip3*phip3);
const double alpha0 = (1.0/35.0)/((beta0 + 1e-10) * (beta0 + 1e-10));
const double alpha1 = (12.0/35.0)/((beta1 + 1e-10) * (beta1 + 1e-10));
const double alpha2 = (18.0/35.0)/((beta2 + 1e-10) * (beta2 + 1e-10));
const double alpha3 = (4.0/35.0)/((beta3 + 1e-10) * (beta3 + 1e-10));
const double alpha_sum_inv = 1.0/(alpha0 + alpha1 + alpha2 + alpha3);
const double w0 = alpha0 * alpha_sum_inv;
const double w1 = alpha1 * alpha_sum_inv;
const double w2 = alpha2 * alpha_sum_inv;
const double w3 = alpha3 * alpha_sum_inv;
return w0 * p0 + w1 * p1 + w2 * p2 + w3 * p3;
}
```
```

**Output:**

**Guide1 Output:** numerical, long, order

**Triggered CoT:** Variable Mapping, Control Block, Values Collection

**Variable Mapping:**

phim3: -8(%rbp), phim2: -16(%rbp), phim1: -24(%rbp), phi: -32(%rbp), phip1: -40(%rbp), phip2: -48(%rbp), phip3: -56(%rbp), p0: -64(%rbp), p1: -72(%rbp), p2: -80(%rbp), p3: -88(%rbp), beta0: -96(%rbp), beta1: -104(%rbp), beta2: -112(%rbp), beta3: -120(%rbp), alpha0: -128(%rbp), alpha1: -136(%rbp), alpha2: -144(%rbp), alpha3: -152(%rbp), alpha_sum_div: -168(%rbp), w0: -176(%rbp), w1: -184(%rbp), w2: -192(%rbp), w3: -200(%rbp)

**Control Block:**

```
double interp_weno7(double phim3, double phim2, double phim1, double
phi, double phip1, double phip2, double phip3){
// part1
const double p0 = (-1.0/4.0)*phim3 + (13.0/12.0) * phim2 + (-
23.0/12.0) * phim1 + (25.0/12.0)*phi;
const double p1 = (1.0/12.0)*phim2 + (-5.0/12.0)*phim1 +
(13.0/12.0)*phi + (1.0/4.0)*phip1;
const double p2 = (-1.0/12.0)*phim1 + (7.0/12.0)*phi +
(7.0/12.0)*phip1 + (-1.0/12.0)*phip2;
const double p3 = (1.0/4.0)*phi + (13.0/12.0)*phip1 + (-
5.0/12.0)*phip2 + (1.0/12.0)*phip3;
// part2
const double beta0 = (phim3*(547.0*phim3 - 3882.0*phim2 +
4642.0*phim1 - 1854.0*phi)
+ phim2*(7043.0*phim2 - 17246.0*phim1 + 7042.0*phi)
+ phim1*(11003.0*phim1 - 9402.0*phi)
+ 2107.0*phi*phi);
const double beta1 =(phim2*(267.0*phim2 - 1642.0*phim1 + 1602.0*phi -
494.0*phip1)
+ phim1*(2843.0*phim1 - 5966.0*phi + 1922.0*phip1)
+ phi*(3443.0*phi - 2522.0*phip1)
+ 547.0*phip1*phip1);
const double beta2 = (phim1*(547.0*phim1 - 2522.0*phi + 1922.0*phip1
- 494.0*phip2)
+ phi*(3443.0*phi -5966.0*phip1 + 1602.0*phip2)
+ phip1*(2843.0*phip1 - 1642.0*phip2)
+ 267.0*phip2* phip2);
const double beta3 = (phi*(2107.0*phi - 9402.0*phip1 + 7042.0*phip2 -
1854.0*phip3)
+ phip1*(11003.0*phip1 - 17246.0*phip2 + 4642.0*phip3)
+ phip2*(7043.0*phip2 - 3882.0*phip3)
+ 547.0*phip3*phip3);
// part3
const double alpha0 = (1.0/35.0)/((beta0 + 1e-10) * (beta0 + 1e-10));
const double alpha1 = (12.0/35.0)/((beta1 + 1e-10) * (beta1 + 1e-10));
const double alpha2 = (18.0/35.0)/((beta2 + 1e-10) * (beta2 + 1e-10));
const double alpha3 = (4.0/35.0)/((beta3 + 1e-10) * (beta3 + 1e-10));
const double alpha_sum_inv = 1.0/(alpha0 + alpha1 + alpha2 + alpha3);
// part4
const double w0 = alpha0 * alpha_sum_inv;
const double w1 = alpha1 * alpha_sum_inv;
const double w2 = alpha2 * alpha_sum_inv;
const double w3 = alpha3 * alpha_sum_inv;
return w0 * p0 + w1 * p1 + w2 * p2 + w3 * p3;
}
```

**LEGO Compilation**

```
#collected values
    .data
.LC_neg_1/4:
    .double -0.25 # -1.0/4.0
.LC_13/12:
    .double 1.0833333 # 13.0/12.0
```

```
.LC_neg_23/12:
    .double -1.916667 # -23.0/12.0
.LC_25/12:
    .double 2.0833333 # 25.0/12.0
```

```
.text
.globl interp_weno7
.type interp_weno7, @function
interp_weno7:
.LFB0:
# Prologue
endbr64
pushq %rbp
movq %rsp, %rbp
subq $480, %rsp
movsd %xmm0, -8(%rbp) # phim3
movsd %xmm1, -16(%rbp) # phim2
movsd %xmm2, -24(%rbp) # phim1
movsd %xmm3, -32(%rbp) # phi
movsd %xmm4, -40(%rbp) # phip1
movsd %xmm5, -48(%rbp) # phip2
movsd %xmm6, -56(%rbp) # phip3
```

```
# Calculate p0
movsd -8(%rbp), %xmm0
movsd .LC_neg_1/4(%rip), %xmm1
mulsd %xmm1, %xmm0
movsd -16(%rbp), %xmm1
movsd .LC_13/12(%rip), %xmm2
mulsd %xmm2, %xmm1
addsd %xmm1, %xmm0
movsd -24(%rbp), %xmm1
movsd .LC_neg_23/12(%rip), %xmm2
mulsd %xmm2, %xmm1
addsd %xmm1, %xmm0
movsd -32(%rbp), %xmm1
movsd .LC_25/12(%rip), %xmm2
mulsd %xmm2, %xmm1
addsd %xmm1, %xmm0
movsd %xmm0, -64(%rbp) # p0
# Calculate p1
# Calculate p2
# Calculate p3
```

| **part2** assembly | **part3** assembly | **part4** assembly |
| --- | --- | --- |

Figure 5: Example workflow for LEGO-Compiler on a full ExeBench example: source code analysis triggers thoughts, including **variable mapping**, splitting **control blocks** and **value collection** illustrated.

# D  METHOD AND EVALUATION DETAILS

This section provides more details figures, tables and further explanations about **LEGO-Compiler** design and experiment evaluation.

## D.1  LEGO-COMPILER: DETAILED DESIGNS

As depicted in Figure 5, LEGO-Compiler is designed to perform a series of thoughts guided by compiler expert knowledge, however, not all CoTs are necessary for each input code, so in our design, we have an **analyze-then-think** approach. First, we will perform an analyzing pass to scan

Table 2: Hardest 10% subset of ExeBench, further breakdown using DeepseekCoder

| Ablation1 | Baseline | Pass@k | Feedback | CoT | LEGO Translation |
|---|---|---|---|---|---|
| DeepseekCoder | 63.5% | 75.5% | 86.0% | 96.5% | 98.5% |
| Ablation2 | Baseline | CoT | Feedback | LEGO Translation | Pass@k |
| DeepseekCoder | 63.5% | 83.5% | 90.5% | 97.0% | 98.5% |

the whole program, whose output flags would trigger necessary Chain-of-Thoughts that will be used in the following process. In this example, the code pattern is majorly about double-precision floating point calculations (**numerical**) and complicated expression evaluation (**order**), besides, the code is too long for direct translation method to handle (**long**). Thus, based on the analysis, we applied the following CoTs:

- **Values collection**: A necessary thought, collecting all variables, numericals in a scanning pass, the **numerical** flag will teach the LLM about assembly knowledge to save numerical values.

- **Variable mapping**: Another necessary thought, which will base on the scanned variables and their types, and form a variable mapping table (SymbolTable) for later compilation.

- **Control Block**: the LEGO translation methodology is applied triggered by **long**, where the entire code is considered too long and will be split into control-block level code snippets via Algorithm 1, it's noteworthy that the **order** flag from analysis will suggest the LLM to split the program into finer-grained blocks so that they can focus more on the order of operations within each block, in Figure 5, there is just one basic block, the flag suggests LLM to split into 4 sequential parts. Then these parts are translated with the aid of SymbolTable individually. Finally, these compiled results are composed together to form a full LEGO compilation.

With different input code, the triggered CoTs will be different, this is helpful because not all thoughts will be useful if no such features appeared in the code, for example, if a code is simple and only has one basic block with a few sequential stmts, then there will be no need to perform LEGO translation related CoTs, because direct translation will be sufficient enough.

D.2 EXEBENCH BREAKDOWN

Figure 3 shows the complexity ablation on the test set of ExeBench, where we use LLM to categorize all cases into three types of complexity based on certain attributes, like code length, expression complexity, control flow complexity, unusual operations occurance, etc. The ablation results show LLMs despite of their models' differences, all get improved on these three categories, where pass@5 and feedback can improve most of the simple cases and some of the medium cases, but can hardly improve on hard cases. While the annotation-based CoT method significantly improve these hard cases, even these mini LLMs can have significant accuracy improvement, except for Codestral model, which fail to follow the CoT correctly, so the result of Codestral for annotation method is a fallback of previous run.

A concern is on whether LLM can categorize code well, so we also perform traditional breakdown, using llvm toolchain (LLVM Project, 2024a) as the frontend analyzer. Based on the analyzed results on basic block count, total instructions and max instructions within a block, we choose the hardest 10% subset of ExeBench for further breakdown. As illustrated in Figure 6, the breakdown characterizes the ExeBench dataset and its hardest 10% subset, which show the subset is significantly harder in total instruction count and basic block count, while the difficulty within each basic block is not significant. After characterization, we use DeepseekCoder as the LLM for evaluation. As depicted in Table 2, although we do find all accuracy degrades due to harder cases, the result further show effectiveness on Feedback, CoT and LEGO translation methods, as the improvement of these methods become more significant.

Furthermore, the hard cases can be majorly categorized into three types:

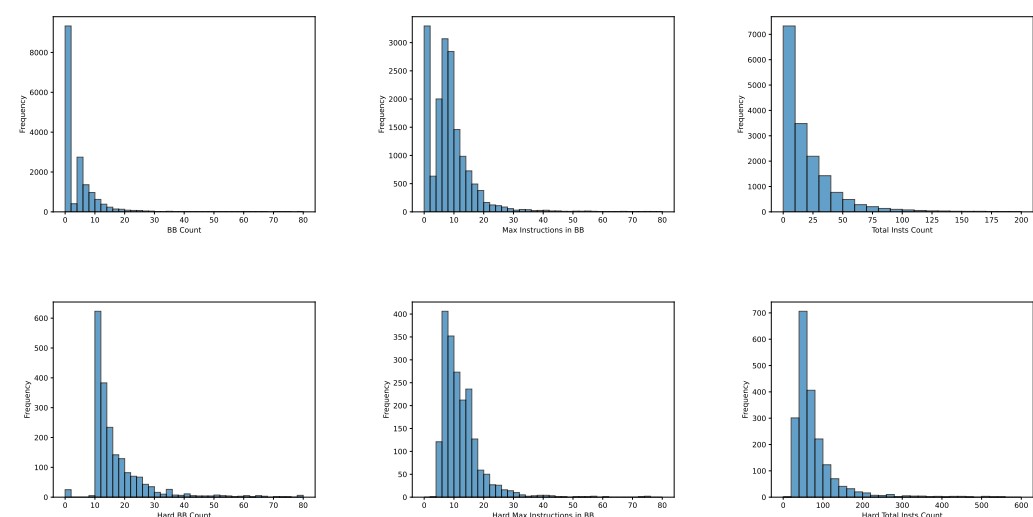

Figure 6: Complexity breakdown of ExeBench and its hard 10% (roughly) subset, we use llvm as the analysis tool, then filter the subset with the following conditions: number of basic blocks(BB) $\geq$ 10 or max instructions in BB $\geq$ 80 or total instructions $\geq$ 200. Upper figures characterize the overall of Exebench and Lower figures characterize the hard 10% subset.

- The insufficiency on some language-specific features, for example, lacking the knowledge of certain operations, which can be definitely improved with more data in the next model pretrained or by providing external knowledge to aid its generation.

- The unsuccessful reasoning during the annotation-based CoTs. This method require the LLMs to reason arithmetic computation and capture specific code patterns in the code to form intermediate results to aid the generation. If the reasoning process generates incorrectly, the CoTs will fail. However, the reasoning capabilities required for this method is not high, majorly the addition and multiplication of integer values within 1000(typically). As LLMs keep improving their abilities in reasoning and math, this type of failures will reduce significantly.

- Very long code reasoning and follow-up generation, where LLMs fail to generate a very large output at once. The first reason is the limitation of current LLMs themselves, although advanced LLMs have increased their context limits into hundreds of thousands tokens, their single generation capability is still limited, to either 4096, 8192 or 16384 tokens. The second reason is the difficulty to generate a long, error-prone output(like assembly languages) at once, this is an intrinsic drawback of direct generation method itself, and can be solved with the proposed LEGO translation/compilation method. LEGO translation can reduce the complexity to control block level, or at maximum, statement level, however, if the statement itself is very long and complicated to evaluate (which is very rare, but potential in modern programming paradigms), our methods will not help, which is a limitation in our work.

## D.3 EVALUATION ON LONGFUNCTION

LongFunction dataset is madeup of 50 C functions in 5 types, where each of them are derived from a certain program pattern like in Figure 7, by alternating the repeated **n**, we could get code size varying from 317 to 238737 tokens, all the token counting is performed by the tiktoken python library, where a gpt-3.5-turbo-0613 vocabulary table is used, although not exactly the token size for each LLM. When evaluating the cases for neural compilation, we compare the neural-compiled results with oracle-compiled results directly since the code is self-contained. As for code translation, we directly test the behavioral output of the translated code and the original code, all compiled by oracle compilers(gcc for C, g++ for C++, CPython runtime for python execution and rustc for

```c
#include <stdio.h>
int arr1[10][10];
typedef struct {
  float f1;
  int i1;
} mystruct1;
typedef struct {
  mystruct1 *s1;
  int i2;
  double d1;
  double d2;
  mystruct1 *s2;
} mystruct2;

void longfunction1(mystruct2 *res) {
  int i, j, k;
  // init
  for (i = 0; i < 10; i++) {
    for (j = 0; j < 10; j++) {
      arr1[i][j] = 0;
    }
  }
  // op1
  for (i = 0; i < 10; i++) {
    for (j = 0; j < 10; j++) {
      for (k = 0; k < res->i2; k++) {
        arr1[i][j] += res->s1->i1;
      }
    }
  }
```
                        Repeat pattern

```c
                            Repeat pattern continue
  // op2
  for (i = 0; i < 10; i++) {
    for (j = 0; j < 10; j++) {
      arr1[i][j] += res->s2->i1;
    }
  }
  // op3
  for (i = 0; i < 10; i++) {
    for (j = 0; j < 10; j++) {
      arr1[i][j] += (int)(res->d1 / res->d2);
    }
  }

// repeat n times(n in 2,4,8,16,32,64,128,256,512)
// will inject a needle in a later repeat pattern
// e.g: += -> -= in op3
  return;
}

int main() {
  mystruct1 s1 = {1.0, 2};
  mystruct1 s2 = {3.0, 4};
  mystruct2 result_struct = {&s1, 3, 18.0, 5.0, &s2};
  longfunction1(&result_struct);
  printf("arr1[0][0] = %d\n", arr1[0][0]);
  // other print
  return 0;
}
```

Figure 7: LongFunction example code: the code is synthesized by repeating certain patterns with n times, and inject a needle in one of the repeating patterns.

rust). It's worth noting that the cases in LongFunction are inspired by needle-in-the-haystack experiment (Kuratov et al., 2024), where a needle in the long context must be correctly picked out. In our LongFunction dataset, this is a small, hard to notice modification of the code pattern, for example, replacing a '+=' with '-='. The ability to identify the needle and translate/compile it correctly could significantly support the LLMs with stronger long-context learning ability.

However, if direct translation/compilation is applied, all the models, despite of their long context limits, fail to translate a near 5k token case, and compile a 2.6k token case in LongFunction, and no need to handle all the above. It's probably LLMs training bias to let it omit similar patterns no matter how we instruct it to step by step thinking and translating.

Our LEGO translation/compilation method, however, can significantly overcome such limitations. Because each time, only a proper sized code snippet is provided to the LLM for further compilation/translation, so theoretically, however long the code is, the LEGO translation method can handle it sufficiently, because small-sized code translation/compilation is assumed to be well-pretrained and proved by results. The splitting and rebuilding processes, although currently not able to be performed all by LLM itself(due to the single output limitation), are simply rule-based and can be well executed by the LEGO-Compiler system, where the splitting process is using the Algorithm 1 algorithm, and the rebuilding process is more simply, concatenating results together.

An easier evaluation can also be performed. By providing an arbitrary code snippet of the long program split following the Algorithm 1, we translate/compile it with the help of globally visible SymbolTable messages and code position markers, if any part of the translation/compilation is semantically correct, then the concatenation of all parts will be correct. This can be easily performed using any LLM api or LLM chat website, and we also provide examples to support this claim.

As a result, all three LLMs (Claude-3.5-sonnet, GPT-4o and DeepseekCoder), successfully translate or compile all the cases in the LongFunction dataset. We also test the capability of newest LLM: o1-preview, although limited to its strict daily usage capacity, it can significantly translate/compile larger sized code snippet, no wonder it can pass all the cases as well.

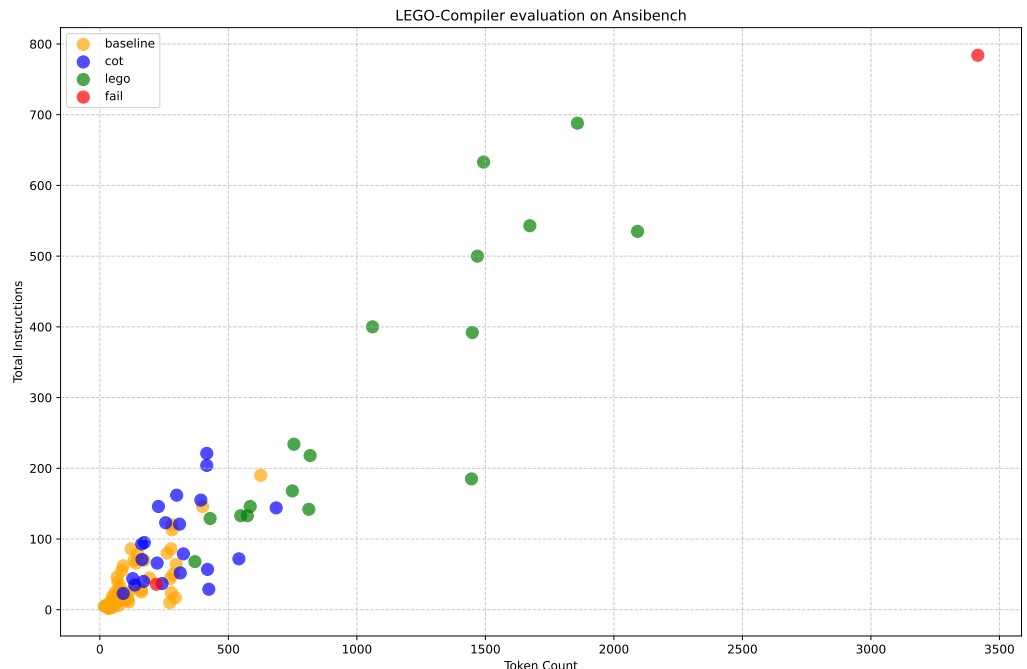

Figure 8: AnsiBench evaluation results using Claude-3.5-Sonnet, the best performant model we evaluated in ExeBench. The **token count** only computes the input length of C code, and typically, the output assembly will be 3-6 times larger in token size.

## D.4 ANSIBENCH: MORE REAL-WORLD CODEBASE EVALUATION

Except CoreMark, we conduct additional real-world codebases evaluation, we use Ansi-Bench (nfinit, 2024), a collection of well-known ANSI C standard benchmark suites (Gustafson & Snell, 1995; Dongarra et al., 2003) besides CoreMark, benchmarking a wide variety of systems and compilers, including a number of classic, industry-standard benchmarks as well as some select programs that can be used as benchmarks.

We evaluate the whole AnsiBench collection with our LEGO-Compiler, using similar evaluation settings of CoreMark. We list the details of every function we compiled in Figure 8, totally we have 96 functions in total, except for a few utility functions that are easy to compile, many of them represents real-world codebase complexity. We ablate the translation methods we applied to showcase both the effectiveness of annotation-based Chain-of-Thoughts and LEGO translation.

LEGO translation method significantly improve the translation scalability of real-world code by near an order of magnitude. In total, we pass 94 out of 96 cases in Ansibench across 7 different codebases, including Whetstone, Dhrystone, Hint(one failure), Linpack, Tripforce(one failure), Stream and CoreMark.

There are majorly three types of errors where the first two types are where LEGO translation out-performs the others significantly.

- The first type is lengthy code input with over a thousand token size (typically), where the output size is truncated by the limits of output model itself. besides, the coarse-grained translation itself is prune to bugs as a simple mistake can cause either compilation error, segmentation fault or silence error. LEGO translation method can significantly reduce such errors, the case in which LEGO translation also fails is the main function of Hint bench-mark, which is more complex than the main function of CoreMark. We analyze its failure, where the reasoning step of the stack allocation fails to generate a correct mapping, there-fore, causing the afterwards failure. Despite this, LEGO translation handles all the other lengthy code correctly as it can breakdown the translation complexity.

```c
static uint8_t func_1(void)
{
  int64_t l_2[1];
  int32_t l_3 = 0xF37831E4L;
  int32_t l_6[3];
  int i;
  for (i = 0; i < 1; i++)
    l_2[i] = 0xEC2E0CF5720E83C7LL;
  for (i = 0; i < 3; i++)
    l_6[i] = 0xA8CDA2AEL;
  for (l_3 = 0; (l_3 >= 0); l_3 -= 1)
  {
    int16_t l_4 = (-1L);
    int32_t l_5 = (-1L);
    int i;
    l_5 = ((l_2[l_3] != 1UL) <= l_4);
    l_6[0] = l_4;
  }
  l_6[2] = l_3;
  return l_6[0];
}
```

```c
struct S0 {
  uint8_t f0;
  int32_t f1;
  uint16_t f2;
};

struct S1 {
  struct S0 f0;
  uint32_t f1;
  struct S0 f2;
  uint16_t f3;
};
static struct S1 func_1(void)
{
  uint32_t l_4 = 0xF054A20AL;
  int32_t l_5 = 0x4B03E386L;
  uint8_t l_6[3];
  struct S1 l_11 = {
    {0x8EL,0x36DC9922L,0xC436L},
    4294967295UL,
    {1UL,0xC3FC0233L,0xD52AL},
    0x2BBDL
  };
  ...
  return l_11;
}
```

Figure 9: Csmith example code, the major body part of the right hand side code is omitted. This example characterizes the necessity of both the Chain-of-Thought reasoning of structs and stack allocation and the LEGO translation method to overcome the complexity of coarse-grained translation.

- The second type of errors is caused possibly by long context forgetting, where the model can not match the current processing assembly with the source code faithfully, LEGO translation method, on the other hand, can handle these cases efficiently as the complexity of each translation is reduced and there are less misleading long contexts to cause these random errors. Besides, finer-grained translation also gives LLMs more attention to faithful translation of operations, the order of operations and implicit conversions.

- The third type is also a limitation our methods can not fully cover: the training bias due to insufficient pretraining in LLMs, which counts for the error in Tripforce's `generate_password` function, where the translation fails to translate the multiple line strings correctly, which is an insufficient training error in Claude-3.5-Sonnet model itself. Another example is, Claude-3.5-Sonnet model is likely to translate the order of the following expression wrongly: `(x - col * 6)`, when it is a postfix of a lengthy expression, it is likely to generate the subtraction instruction first then the multiplication (causing failures), which is not the case for GPT-4o model and Deepseek model. However, for these models, they have more other training bias that make themselves worse than Claude-3.5-Sonnet model. Using Pass@k and feedback correction can mitigate such failures. Besides, we can be positive about these failures because as LLMs advance, these failures will gradually disappear.

### D.5  CSMITH: RANDOMLY GENERATED PROGRAMS EVALUATION

Except for AnsiBench evaluation. We further perform evaluations on randomly generate programs with sufficient complexity. We use Csmith (Yang et al., 2011), a random generator of C programs which is widely used for finding compiler bugs using differential testing as the test oracle. Typically, Csmith examines compilers with random programs with corner case features and numbers, testing the robustness of compilers. Code examples generated from Csmith are illustrated in Figure 9.

As depicted in Figure 10, randomly generated programs by Csmith are very hard for both baseline and CoT-only methods to translate. In a test suite of 25 cases LEGO translation successfully pass, we find baseline translation can only pass 4 cases, with CoT translation, only 9 more cases can be passed. Besides, the complexity of cases only passed by LEGO translation method are signif-

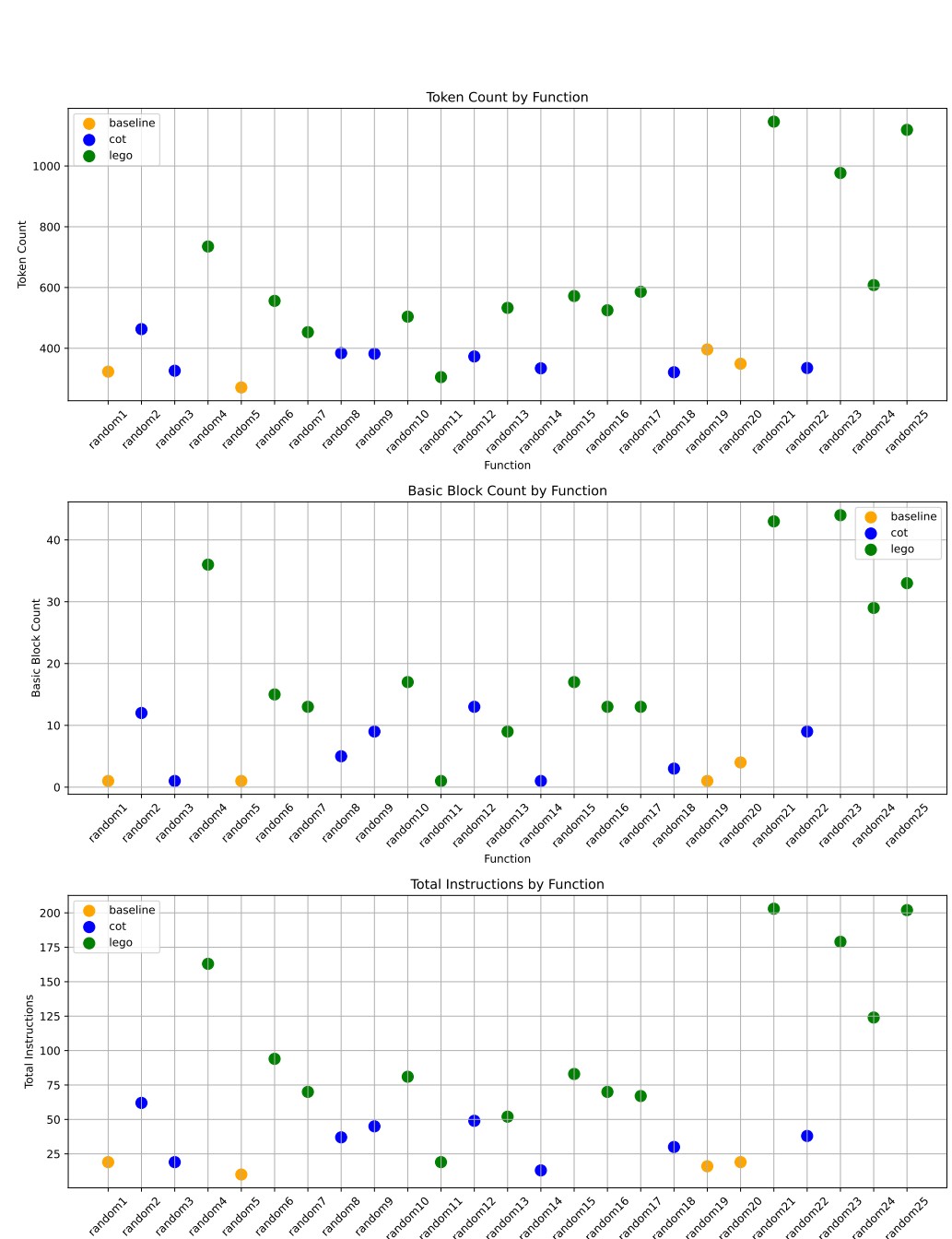

Figure 10: Csmith random generated code statistics, where the practical utility of the LEGO method is show clearly by passing significantly more complex cases.

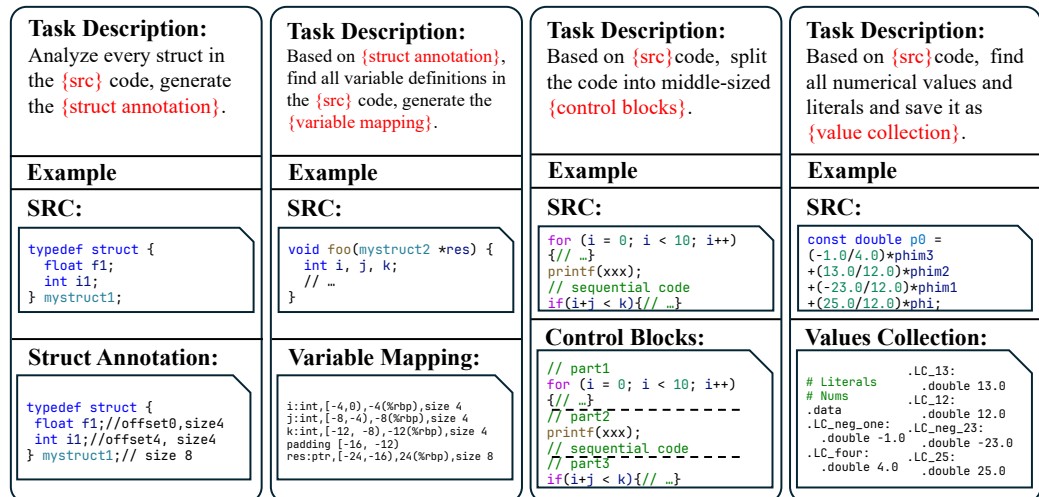

Figure 11: Annotation-based Chain-of-Thoughts prompts for neural compilation

Table 3: Ablation study: impact of temperature on Pass@1 and Pass@5 performance

| Model | Pass@1 | | | | | Pass@5 | | | | |
|---|---|---|---|---|---|---|---|---|---|---|
| | 0.2 | 0.4 | 0.6 | 0.8 | 1.0 | 0.2 | 0.4 | 0.6 | 0.8 | 1.0 |
| GPT-4o | 71% | **73%** | 72% | 72% | 72% | 79% | 83% | 86% | 89% | **92%** |
| Claude-3.5-Sonnet | 87% | 91% | **93%** | 88% | 89% | 91% | 92% | **96%** | 94% | **96%** |
| DeepseekCoder | **89%** | 88% | 86% | 87% | 88% | 92% | 92% | 92% | **93%** | 92% |
| GPT-4o-mini | **64%** | 61% | 61% | 60% | 60% | 71% | 71% | 79% | 73% | **80%** |
| Claude-3-Haiku | **79%** | 76% | 78% | 72% | 73% | 82% | 84% | 85% | **86%** | **86%** |
| Codestral | **73%** | 66% | 41% | - | - | 84% | **90%** | 73% | - | - |

icantly larger than others, which can be characterized by token count, basic block count and total instructions in the three subfigures respectively.

During Csmith evaluation, we also identify several kinds of errors during LEGO-Compiler translation. For example, overflow value assignment is a kind of error which doesn't usually occur in daily programming but can be found during compiler testing. Taking `int16_t x = 0x56671485;` as an example, it will trigger errors because LLMs directly generate `movw $0x56671485, x`'s `address`, which fails to check whether the value (overflows the 16 bit word) can be represented through `movw` instruction. Another example is, when handling with implicit type conversions, LLMs may not promote the type correctly, this is critical for floating point computation as operations with wrong precision will cause numerical errors.

## D.6 Other evaluation details

Table 3 shows the impact of temperature when using LLMs for neural compilation. LLMs have better Pass@1 accuracy when temperature is low, but higher Pass@5 accuracy when temperature is high. This is as expected, since temperature influences the decoding process, with higher temperature, the results are more diverse, allowing LLMs to jump out of pretraining bias, however, this could also cause more errors by choosing sub-optimal decoding tokens that may cause errors.

Figure 11 explains how we prompt LLMs to do the annotation-based Chain-of-Thoughts to aid the neural compilation process.

# E LIMITATIONS

**Optimization Capabilities**: The current focus of LEGO-Compiler is on functional correctness rather than code optimization. Traditional compilers excel at producing highly optimized code, a capability not yet matched by our neural approach. Future work could explore integrating optimization techniques into the neural compilation process.

**Performance Overhead**: As noted in the discussion, the computational cost of neural compilation is significantly higher than traditional methods. This limitation may restrict its practical application in scenarios where compilation speed is critical.

**Complex Expression Handling**: The paper acknowledges challenges in managing highly complex expressions, proposing external tool integration or expression decomposition as potential solutions. This indicates a current limitation in LLMs' ability to handle intricate code structures independently.

**Architecture-Specific Knowledge**: While the paper demonstrates success with x86, ARM, and RISC-V architectures, expanding to a broader range of architectures, especially more specialized ones, may require significant additional training or fine-tuning of the LLMs, or by providing large RAG database to provide such knowledge in the context.

**Security and Reliability**: The stochastic nature of LLM outputs raises concerns about the consistency and security of the generated assembly code. Ensuring deterministic outputs and preventing potential vulnerabilities introduced by the neural compilation process remains a challenge.

**Handling of Language-Specific Features**: The paper primarily focuses on C-like language compilation, and proves the availability of functionality in neural compilation through both theoretical and empirical results. However, extending the approach to other programming languages can result in more tailored problems, for example:

- **RAII idiom**: Languages with class properties, like C++, have an important programming idiom called **R**esource **A**cquisition **I**s **I**nitialization(**RAII**), which pose significant challenges for LLMs. For instance, constructor and destructor functions in these languages are implicitly called based on scope. This implicit behavior is difficult for LLMs to accurately model and implement in assembly code, but this could be solved using external mangling tools like **c++filt** (Free Software Foundation, 2023).

- **Name Mangling**:Languages like C++ and Rust use name mangling mechanisms for function overloading and template instantiation. This requires special handling of global symbols such as function names during compilation, which may be challenging for LLMs to consistently implement without explicit training on these concepts.

- **Dynamic Language Features**: Some language features violate the composability principle that LEGO translation relies on. For example, Python's exception handling mechanism, which can cross scope boundaries, would make the LEGO translation method ineffective for such features.

It's important to note that many of these challenges are not unique to neural compilation. Traditional compilers also struggle with highly dynamic features like exception handling and Run-Time Type Information (RTTI). Languages like Python achieve their flexibility by sacrificing native code generation in favor of interpretation or JIT compilation. Therefore, these limitations are not specific to our work but rather inherent to any approach based on static compilation analysis.

The ability to handle these diverse language features represents an area for future research in neural compilation. It may require developing specialized techniques or combining neural methods with traditional compiler approaches to address these complex language-specific challenges.

Scalability to Very Large Codebases: While the LEGO translation method significantly improves scalability, handling entire large-scale software projects or operating systems may still be beyond the current capabilities of this approach. However, It is noteworthy that repository complexity is naturally reduced into files or functions, therefore, LLM-based compilers and translators are potential to translate them with more advanced models and more carefully designed methods.

