# OpenReview forum: "LEGO-Compiler: Enhancing Neural Compilation Through Composable Chain of Thought"
_ICLR.cc/2025/Conference — Submitted to ICLR 2025_

### Official Review · Reviewer_wpuL · 2024-11-03

**Soundness:** 2
**Presentation:** 3
**Contribution:** 2
**Rating:** 6
**Confidence:** 4

**Summary:**

This paper targets at an important problem of neural compilation, which translates high-level source code into assembly code with Large Language Models (LLMs). The authors propose a new method called LEGO-Compiler, which decomposes a large high-level program into small trunks, compiles individual code snippets, and then composes them together. The authors demonstrate that the LEGO-Compiler achieves an accuracy of over 99% on ExeBench and nearly 100% on CoreMark, which are two benchmarks for evaluating neural compilations.

**Strengths:**

1. Targeting an important problem of neural compilation.
2. Novel method that adopts the idea of decomposing and composing to handle large complex programs.
3. Proofs of the correctness of the LEGO approach and clearly stated assumptions and limitations.

**Weaknesses:**

1. Weak benchmarks: the evaluated benchmarks are not complex enough to demonstrate the utility of the LEGO-Compiler.

The program sizes of the benchmarks can be too small. Figure 6 shows the complexity of the ExeBench. The number of programs whose total instruction count exceeds 100 is scarce. Even for the top 10% of most complex programs, the total instruction count hardly exceeds 200. Considering that a statement line in the high-level language may be translated into multiple assembly instructions, the ExeBench could contain only a few dozen lines of code, which is too small to demonstrate the utility of the LEGO-Compiler. Although the authors also evaluate a more complex benchmark, the LongFunction benchmark, the number of complex programs is not reported.

The language features and library usage in the benchmarks are limited. One of the challenges in compilation is the handling of various language features and library calls. For example, multi-dimensional arrays and pointers to functions can be challenging to compile correctly. However, from the examples given in the paper, the benchmarks only contain simple pointer operations and arithmetic operations, which are not complex language features. Moreover, library calls can be hard to compile. System calls, such as `fork` and `execve`, require special handling in the assembly code. The authors do not discuss how the LEGO-Compiler handles these system calls. It is unclear whether the evaluated benchmarks contain such system calls.

Without an understanding of the complexity of the benchmark, it can be hard to draw a conclusion on the effectiveness of the LEGO-Compiler.

2. Lack of comparison with other state-of-the-art methods.

The authors mentioned multiple pieces of work that also target the problem of neural compilation. However, none of the compared approached is evaluated in the paper. The authors should compare the LEGO-Compiler with other state-of-the-art methods to demonstrate the effectiveness of the LEGO-Compiler. In particular, the authors should show the improvement of the LEGO-Compiler over the state-of-the-art methods. Without such comparison, the advantage of the LEGO-Compiler is unclear.

3. Insufficient correctness evaluation on the generated assembly code.

The authors do not verify the correctness of the generated assembly code. The authors mention that they run the test cases provided by the benchmarks, and check whether the output of the generated assembly code passes the test cases. However, this does guarantee that the generated assembly code is correct. Testing cases can be incomplete, and may not cover all behaviors of the program. It can be the case that the generated assembly code happens to pass all the test cases, but is still non-equivalent to the original high-level program. The authors should verify the equivalence of the generated assembly code with the original high-level program. SMT solvers can be a useful tool for this purpose.

Without the correctness verification on the generated code, the achieved accuracy of the LEGO-Compiler is not convincing.

4. The improvement with the LEGO-Compiler compared to baselines is not significant.

The improvement of the LEGO-Compiler over the baseline is less than 1% on the ExeBench, which is not significant. As shown in Table 1, even without the LEGO-style approach, the baselines can already achieve over 98% accuracy. LEGO-Compiler only improves by 1% compared to the Chain-of-Thought baseline. The insignificant improvement raises the question of whether the LEGO-Compiler is necessary.

Moreover, the accuracy of the LEGO-Compiler for all the small LLMs is missing. Also, the Chain-of-Thought baseline is missing for Codestral. The reason behind such missing experiments is not clear. What are the limitations of applying LEGO-Compiler to small LLMs? Answering this question can help understand the limitations and application scope of the LEGO-Compiler.

**Questions:**

1. How does the LEGO-Compiler handle complex language features and library calls?
2. How does the LEGO-Compiler perform on benchmarks with over 1000 lines of code?
3. How does the LEGO-Compiler compare with other state-of-the-art methods on neural compilation?
4. What is the true accuracy after verifying the correctness of the generated assembly code?
5. Why is the improvement of the LEGO-Compiler over the baseline insignificant?
6. What are the limitations of applying LEGO-Compiler to small LLMs?

---

> ### Author Response · Authors · 2024-11-17
> **Response to reviewer wpuL (Part 1 of 2)**
>
> Dear Reviewer wpuL:
>
> Thank you for your thorough and thoughtful review. Below, we have provided detailed responses to the identified weaknesses and questions, with similar points grouped together for clarity.
>
> # 1. Weak benchmarks: the evaluated benchmarks are not complex enough to demonstrate the utility of the LEGO-Compiler.
> To clarify, CoreMark is not a toy benchmark. Regarding ExeBench, although it is not very complex, it is a benchmark widely used for neural compilation tasks. For more details, please refer to the **1st response in the general reply**. In short, we are actually pioneers in scaling up the capability of current LLMs for both neural compilation and neural code translation tasks. While the complexity of ExeBench itself is somewhat limited, we evaluate it primarily for fair comparison to related works, which also use ExeBench for evaluation.
>
> Although ExeBench is not complicated in length, it covers almost all C language features, such as system calls and inline assembly usage, Achieving high accuracy on ExeBench is a foundation for evaluating more complex cases. Therefore, we also evaluate our LEGO-Compiler with a case study of CoreMark, which is significantly more complex and has been acknowledged as such by other reviewers.
> Additionally, we introduce the LongFunction benchmark to demonstrate translation scalability, which we explain in detail in the response to **Question 6**.
>
> To further address your concerns, we are currently conducting experiments on more real-world C codebases to showcase the utility of LEGO-Compiler. We will update the results once the evaluation is completed!
> # 2. The language features and library usage in the benchmarks are limited. System calls, such as fork and execve, require special handling in the assembly code. How does the LEGO-Compiler handle complex language features and library calls?
> There seems to be a misunderstanding here. System calls and library calls are simply function calls from the compiler's perspective (including LEGO-Compiler). In the resulting assembly output, they are just calls with some additional calling conventions (for syscalls). We have not encountered significant issues caused by library function calls, as it is the programmer’s responsibility to handle the return values of these functions. Most errors we identified are analyzed in **Appendix D.2**, the ExeBench breakdown section.
> As for complex language features, some do require special handling, like the one we did for struct, union and array should be similar. Many features can be handled well by LLMs due to their pretraining, such as function calling ABIs, instruction-level translation, pointer operations, and indirect jumps. However, there are features with which LLMs still struggle. For example, handling expressions with very high depth, like **a+(b*d/(c-e))/(f+g)-h**, or architecture-specific knowledge like AVX intrinsics, poses challenges. We have discussed them in **Appendix E**, the limitation section.
> The link below is an example of how fork-exec example code can be neural-compiled correctly with CoTs, which can be easily reproduced:
>
> ***https://www.perplexity.ai/search/i-want-you-to-act-like-a-compi-OPQbOBCXQkq2e2TvTwzuGA***
>
> # 3. Lack of comparison with other state-of-the-art methods.
> Answer: Please refer to our **3rd response in the general reply**, where smaller non-LLM neural compilers are performing worse than foundation LLMs, therefore we think it’s not necessary to compare with. We will later update our paper to include direct comparisons with Zhang, et.al, which wasn't available during our initial submission.

---

> ### Author Response · Authors · 2024-11-17
> **Response to reviewer wpuL (Part 2 of 2)**
>
> # 4. Without an understanding of the complexity of the benchmark, it can be hard to draw a conclusion on the effectiveness of the LEGO-Compiler. How does the LEGO-Compiler perform on benchmarks with over 1000 lines of code?
> To address this question, first, as illustrated in **Figure 4**, the main function of CoreMark serves as a representative example with 2092 tokens, 215 LOCs, 48 basic blocks, and 535 instructions. This complexity surpasses the capabilities of all previous work; even our method without LEGO translation cannot compile it. Therefore, the effectiveness of LEGO translation is demonstrated by its capability to handle long and complex functions with lengthy code patterns.
>
> For the question on how LEGO-Compiler performs on benchmarks with over 1000 lines of code, we need to clarify two things:
>
> - 1)If the question refers to benchmarks with over 1000 lines of code per file or repository, CoreMark already qualifies, as it contains over 1000 lines of code across 40 functions, all of which are compiled correctly by LEGO-Compiler. We are also conducting further experiments on more real-world codebases, as mentioned in **Question 1**.
>
> - 2)If the question refers to functions with over 1000 lines of code, it’s important to note that functions of this length are very rare in modern programming paradigms. Besides, we also conduct lengthy function experiments with LongFunction dataset that can be up to 37k LOC, which has shown that LEGO translation can reduce the complexity of function-level translation into finer granularity, which is the core contribution of our methods. The LongFunction dataset is provided in the supplementary materials, within the datasets folder.
> # 5. Insufficient correctness evaluation on the generated assembly code, think we need SMT solver to guarantee full translation equivalence.
> Please refer to our **4th response in the general reply**.
> # 6. The improvement with the LEGO-Compiler compared to baselines is not significant. Why is the improvement of the LEGO-Compiler over the baseline insignificant?
> We would like to clarify that the LEGO translation method does not appear significant in ExeBench evaluation because the former CoT and error-feedback methods are able to handle most of the complexity in ExeBench, and it shows power when the code length is relatively long. However, calling this improvement insignificant is not fair, as the prior methods have already pushed the baselines to very high levels. We present the results on **harder cases** and show more significance for both CoT and LEGO translation methods in **Table 2**.
>
> Another important point is that improvements in neural compilation capabilities are not linearly reflected in accuracy metrics, as different code cases have varying difficulties in compiling correctly. Therefore, it is more meaningful to assess in another view: **how complicated cases can LEGO-Compiler translate correctly**. If LEGO-Compiler can translate a very complicated case, LEGO-Compiler can translate simpler similar cases with high possibility. From this perspective, we believe the improvement is indeed significant.
>
> Additionally, the significance of LEGO translation method is majorly tested through LongFunction evaluation, where we find that without the LEGO translation method, code translation task fails at 5k token input and compilation task fails at 2.6k token, even with powerful o1-preview model. However, equipped with the LEGO translation method, all code translation and compilation cases are passed, where the maximum function length can be over 238k. Although it is more like a needle-in-the-haystack experiment because the code patterns are simple in LongFunction, and each part translation is easy and somehow redundant, it showcases the scalability of LEGO translation because direct translation method can not handle such lengthy input. LEGO translation, on the other hand,  significantly reduces the translation complexity and can handle lengthy translation. The details are provided in **Appendix D.3**.
> # 7. Missing results for small LLMs and codestral.What are the limitations of applying LEGO-Compiler to small LLMs?
> In our scenario, LLMs need to follow instructions provided in the prompt to perform complex tasks such as CoT translation and LEGO translation. We have observed that smaller LLMs struggle to follow the instructions required for LEGO translation. We are continuing to refine the LEGO translation framework to make it more robust and applicable across different LLMs. This is why we have not fully tested smaller LLMs.
>
> As for Codestral, the situation is similar. It has difficulty following instructions in the CoT method, resulting in worse CoT outcomes compared to its pass@5 + feedback results.
>
> Finally, thank you again for your detailed and thoughtful review. We hope the above responses address most of your concerns.
>
> Sincerely,
>
> Submission931 Authors

---

> > ### Comment · Reviewer_wpuL · 2024-11-20
> >
> > > 1. ...we are currently conducting experiments on more real-world C codebases...
> >
> > I agree that this experiment is essential for showcasing the practical utility of the LEGO method.
> >
> > > 6. ...the significance of LEGO translation method is majorly tested through LongFunction evaluation...
> >
> > LongFunction seems to lack diversity, as it only shows the needle-in-the-haystack pattern. It only shows that the LEGO method outperforms the baseline approaches in this particular pattern. I have two suggestions:
> > 1. Conduct ablation studies on real-world C programs with similar complexity to LongFunction
> > 2. Randomly generate programs with sufficient complexity and show the improvement of the LEGO method against baselines.

---

> > > ### Author Response · Authors · 2024-11-22
> > > **Update with new evaluation results**
> > >
> > > Dear Reviewer wpuL:
> > >
> > > We have currently updated our manuscript and conducted both more real-world C program evaluation and randomly generated program evaluation, with method ablations. We have updated the results in Appendix D.4 and Appendix D.5.
> > >
> > > We hope the above updates address your concerns. In short, we find the LEGO translation method significant as illustrated in Figure 8 and Figure 9, where the LEGO translation method can scale up the translation by near one order of magnitude.
> > >
> > > **If our rebuttal better aligns with your expectations,we kindly invite you to reconsider your initial rating and soundness.**
> > >
> > > Sincerely,
> > >
> > > Submission931 authors

---

> > > > ### Comment · Reviewer_wpuL · 2024-11-22
> > > >
> > > > Some additional questions about the evaluation of Csmith:
> > > >
> > > > 1. How many test cases are generated?
> > > > 2. What settings do you use when generating programs with Csmith?
> > > > 3. How many test cases does the LEGO compiler pass in total?

---

> ### Author Response · Authors · 2024-11-22
> **Reply to questions regarding Csmith**
>
> ### How many test cases are generated?
> We actually try a bunch of different settings as we are also learning the usage of Csmith with your suggestions to evaluate with randomly generated code. In our current version, we generate 40 cases with the setting described in question2.
> ### What settings do you use when generating programs with Csmith?
> We use the following settings in the results of Figure 9 and Figure 10:
>
> *csmith --quiet --concise --no-volatile-pointers --no-builtins --no-volatiles --no-unions --no-checksum --no-jumps --no-bitfields --no-packed-struct --no-safe-math --no-global-variables  --max-expr-complexity 2 --max-block-depth 3 --max-block-size 2 --max-funcs 1 --max-array-dim 2 --max-array-len-per-dim 10*
>
> We majorly limit settings, like union, global variables, volatile pointers, because Csmith tends to make the code significantly harder which is way beyond how well-formed C code is usually written.
> For example, Csmith will generate nested structs with unions and unions with structs, which can cause the reasoning in CoT fail and therefore, cannot help us evaluate the effectiveness of LEGO translation.
>
> ### How many test cases does the LEGO compiler pass in total?
> We pass 25 out of 40 cases, where all cases are generated in the same settings described in 2. The code generated by Csmith can vary hugely in its complexity as depicted in Figure 10. The failed cases are not presented in Figure 10 as the purpose of Csmith evaluation is to showcase the effectiveness of LEGO translation. Besides, the failures are mostly not caused by LEGO translation but the complexity of statement level translation. However, it is helpful as it does help us to reveal the challenging edge cases and patterns LLMs struggle to handle, like the assignment of overflow values (we discussed in Appendix D.5, line 1764-1771).

---

> > ### Comment · Reviewer_wpuL · 2024-11-25
> >
> > The evaluation results on more real-world programs look promising.
> >
> > The results on Csmith are not so positive (only 62.5% of test cases are passed). Nonetheless, the analysis of the failure cases is insightful and convincing. Also, Csmith-generated test cases do not commonly appear in real-world usages.
> >
> > Based on the reasons above, I would be happy to revise my rate on this paper.

---

> > > ### Author Response · Authors · 2024-11-26
> > >
> > > Dear Reviwer wpuL,
> > >
> > > Thank you very much for your helpful feedback and for acknowledging the improvements made during the discussion period.
> > >
> > > Submission931 Authors

---

### Official Review · Reviewer_vsfE · 2024-11-03

**Soundness:** 3
**Presentation:** 3
**Contribution:** 2
**Rating:** 5
**Confidence:** 3

**Summary:**

The paper proposes to use a set of methods to improve the quality of neural code translation. More precisely, the authors propose three methods: 1) LEGO translation, which decomposes large programs into smaller blocks; 2) Using annotations and Chain-of-Thought to guide the LLM through the compilation process; and 3) Using feedback obtained after compiling the translated code for self-correction. The authors evaluate their proposed method on a subset of 500 C codes randomly selected from ExeBench (a dataset of 40k executable C programs). They show that their proposed method improves the quality of code generated by existing LLMs when evaluated on the proposed benchmark (GPT-4o, Claude-3.5-Sonnet and DeepseekCoder). Claude-3.5-Sonnet improves by 7.4% while GPT-4o improves by 23%.

**Strengths:**

-	The problem of Neural Compilation is hard and challenging. A minor mistake can make the generated code invalid.
-	Achieving an accuracy reaching 99% and even 100% on the task of neural compilation is an interesting result.

**Weaknesses:**

While the proposed work is important, and while the results are interesting, I do not believe that the contribution is strong enough. One of my concerns with the paper is that the authors combine a set of well-known and obvious methods and apply them in the context of neural compilation to improve its quality. There is clearly a degree of novelty in doing this, but I believe it is not strong enough. This said, I still think the work and the results are quite interesting.

One of the main ideas proposed in the paper is to compile code block by block instead of compiling it in one shot. The authors do not provide a clear ablation study showing the effect of this idea (alone). I would suggest adding an experiment where you take the baseline and use it to compile code block-by-block (without using the pass@k method, where you reject the generated code if it does not compiler and generate other samples of the code). The pass@k column in Table 1 currently includes two techniques: the translation block-by-block and the detection of syntactic errors in the generated code. Can you add a column to evaluate the use of block-by-block compilation when applied alone?

The authors do not provide a detailed analysis of how long their compilation process takes. They propose an iterative process which might take a long time and therefore it is important to evaluate how long compilation takes.

The current paper also does not compare with previous work. While such a comparison is not necessary, it would have helped the reader better appreciate the contribution of this paper. For example, the authors could have generated LLVM-IR using LLMs and compared with Guo and Moses [1]. I’m saying this because adapting your tool-chain to generate LLVM-IR does not require a significant amount of work since you already generate code for x86 64, arm-v8a, and riscv64. This said, I do not think this experiment is essential for the validation of your contribution. It is rather recommended.

It would be interesting to show a breakdown of the accuracy of the LEGO-compiler on the different hardware architectures (x86 64, arm-v8a, and riscv64). Does the LEGO-compiler do well on all of them?

[1]  Zifan Carl Guo and William S. Moses. Enabling transformers to understand low-level programs. In 2022 IEEE High Performance Extreme Computing Conference (HPEC), pp. 1–9, 2022. doi:10.1109/HPEC55821.2022.9926313.

**Questions:**

-	Can you add an ablation study to evaluate the use of block-by-block compilation alone without rejecting code that does not compile?
-	Can you evaluate how long it takes for the LEGO-compiler to compiler code?
-	I recommend adding a comparison with Guo and Moses (2022). While this experiment is not necessary, it will better clarify the strength of your contribution.

---

> ### Author Response · Authors · 2024-11-17
> **Response to reviewer vsfE (Part 1 of 2)**
>
> Dear Reviewer vsfE:
>
> Thank you for your in-depth review. We appreiciate the time you spent assessing our work. Please find our responses to the weaknesses and questions below:
> # 1. Contribution concern:
> We believe there may be some misunderstandings regarding our contributions. Our work presents three key novel points: the **use of CoTs in neural compilation**, the **implementation of error feedback** mechanisms, and most importantly, we are **the first to reveal and leverage the composable nature of code translation**, proposing the LEGO translation method which can be applicable **not only to neural compilation but to broader code translation tasks**. It requires theoretical support to translate in finer granularity and then combine the results together, which we formalize as translation composability. We believe this is a significant novel contribution that can benefit the broader code translation community, enabling many LLM-based code translation tasks to scale up for production-level usage.
> # 2. Standalone ablation study on LEGO translation method:
> To clarify, the pass@k column in **Table 1** only contains k times generation, and the block-by-block translation is applied only on the LEGO-Compiler column in **Table 1**, after CoT.
> Your suggestion on using different ablation orders when applying methods is helpful.
> Actually, ExeBench may not be complex enough to showcase the capability of our block-by-block LEGO translation method.  Therefore we try to showcase the LEGO translation capability through the evaluation on the LongFunction benchmark we proposed. In this benchmark, without the LEGO translation method, code translation fails at 5k token input, and compilation fails at 2.6k tokens, even with powerful models like o1-preview. However, when equipped with the LEGO translation method, all code translation and compilation cases pass, with the maximum function length exceeding 238k tokens. Although this may seem like a "needle-in-the-haystack" experiment due to the simple and somewhat redundant code patterns in LongFunction, it demonstrates the scalability of LEGO translation. The method significantly reduces translation complexity and can be combined to form a full, lengthy translation. Details can be found in **Appendix D.3**.
> Additionally, we are currently extending our experiments to broader real-world applications and will ablate LEGO translation alone.
> # 3. Comparison with previous work:
> Please refer to our **3rd response in the general reply**, where smaller non-LLM neural compilers are performing worse than foundation LLMs, therefore we think it’s not necessary to compare with.
>
> As for comparison with Guo, et.al in C-LLVM IR experiments, we think the comparison with them using exactly matched setup is not necessary.  However, we will conduct such experiments later by testing the meaningful C-LLVM IR generation results with ExeBench.

---

> > ### Author Response · Authors · 2024-11-30
> >
> > Dear reviewer vsfE :
> >
> > We would like to remind you that the discussion period is ending, and we haven’t heard from your responses. We have added several important experiments and have already addressed reviewer wpuL’s concerns.
> >
> > We would really appreciate it if you could spent some time to review the rebuttal we have updated and update your review. We will be happy to clarify any questions related.
> >
> > Sincerely,
> >
> > Submission931 authors

---

> ### Author Response · Authors · 2024-11-17
> **Response to reviewer vsfE (Part 2 of 2)**
>
> # 4. Compilation time analysis:
> We acknowledge the importance of evaluating compilation time performance. Let us provide a quantitative analysis using a representative example. For a source program of 200 tokens, traditional compilers like GCC complete compilation in approximately 20ms. In contrast, our LLM-based approach using Llama-3.1-8B (deployed with vllm on an A100 80GB GPU) generates approximately 600 tokens of assembly output at a rate of 40 ms per token, resulting in a total processing time of about 20 seconds. For commercial LLMs like GPT-4o and Claude 3.5, the process is more effective, allowing them to compile code more quickly. The performance of function-level code compilation for these models is typically at the second level, which is 2-3 orders of magnitude slower than traditional compilers like GCC.
>
> While this computational overhead is substantial, it aligns with the expected performance characteristics of decoder-only LLMs in translation tasks. Our primary objective is to explore and understand the fundamental capabilities of LLMs in program translation, rather than to compete with the efficiency of traditional compilers.
>
> The use of iterative processes like pass@k and pass@k@fix_round and the inclusion of Chain-of-Thought reasoning could further increase the output token count and processing time. We now report the time analysis in ExeBench using GPT-4o:
> | **Stage**                   | **Remaining Cases** | **Time Taken** | **Average Time per Case** | **Cases Passed** |
> |-----------------------------|---------------------|----------------|---------------------------|------------------|
> | Baseline generation          | 500                 | 24 minutes     | 3.75 s/case                | 384         |
> | pass@5 generation            | 116                 | 40 minutes     | 20.69 s/case               | 83        |
> | 3 self-correction rounds     | 33                  | 38 minutes     | 69.1 s/case                | 22           |
> | CoT generation               | 11                  | 37 minutes     | 201 s/case                 | 7                |
> | LEGO translation method      | 4                   | 21 minutes     | 315 s/case                 | 3          |
> |Overall                          | 500                    |  160 minutes | 19.2 s/case    |      499  |
>
> In conclusion, the average pass time across all cases was 19.2 seconds per case, showing that our methodology is generally efficient, especially considering the complexity of neural compilation tasks. While the more advanced techniques, such as CoT and LEGO translation, took more time per case, they were necessary only for a small subset of challenging cases, and the majority of cases were handled swiftly. We will later reflect the above analysis to the manuscript.
> # 5. More detailed results on different hardware architectures:
> Performing full scale results on different hardware architectures would require substantial additional work, as the majority of our evaluation is based on x86_64. However, we have performed ARM and RISC-V experiments on the LongFunction benchmark, where the architecture related complexity is not influencing the results and the LEGO translation method performs all well on them. As for detailed results on ARM and RISC-V with more complicated cases like in ExeBench and CoreMark, we do find some interesting findings. For example, there are new semantic challenges like the restricted bit width of immediate values, so not all numbers can be expressed in the way the current model generates, and the calling convention of different architectures will cause some related errors as well.
>
> We hope the above responses help address your concerns. Thank you again for your valuable review!
>
> Sincerely,
>
> Submission931 Authors

---

> ### Author Response · Authors · 2024-11-22
> **Update with new evaluation results**
>
> Dear reviewer vsfE:
>
> We have now updated our manuscript and conducted more real-world C program evaluation and randomly generated program evaluation, with method ablations. We have updated the results in Appendix D.4 and Appendix D.5.
>
> The results show the effectiveness of LEGO translation method as it scales up the complexity of code which LLMs are capable to compile by near an order of magnitude. We revise the ablation designs, where we think both the pass@k and the n round self-correction with error feedback are applicable to direct translation, CoT translation and LEGO translation, as they are internal capabilities within LLMs themselves, so we think they are better to be applied.
>
> We hope the above updates address your concerns. **If it is the case, we kindly invite you to reconsider your initial rating.**
>
> Sincerely,
>
> Submission931 authors

---

### Official Review · Reviewer_Mneu · 2024-11-03

**Soundness:** 3
**Presentation:** 3
**Contribution:** 2
**Rating:** 5
**Confidence:** 5

**Summary:**

The paper presents a technique for neural translation from high-level language to assembly code, focusing on functionality (and not optimization). The core idea is to decompose the program into basic blocks (following traditional compilers) and perform non-optimized translation of each basic block using an LLM. To maintain a mapping of variables to low-level memory locations, LEGO-Compiler uses a separate LLM step. The memory mapping is used as context for the translation of each basic block. To deal with identical variable names across scopes, LEGO-Compiler applies a renaming preprocessing pass. The translation is shown to be effective on 500 programs from ExeBench and additional examples from CoreMark.

**Strengths:**

- The first work I'm aware of to show effective translation of reasonably sized programs
- Compelling approach:
	- decompose large programs into basic blocks
	- annotation-based CoT for providing memory mapping
	- feedback mechanism for self-correction

**Weaknesses:**

- What is the motivation for this work? You seem to replace the parts of the compiler that are computationally more efficient with syntax-directed translation (SDT), and suggest no benefit in doing that. I am sure you have something in mind, so please make it more explicit.
- 99% accuracy on a testset of 500 examples from ExeBench is very different from the "over 99% on ExeBench" (40k examples, or 23k examples after filtering). A different claim at a factor of x46.

**Questions:**

(1) what is the motivation for this work?

- Many steps in LEGO-Compiler are very similar to the ones applied by a traditional compiler.
	- values collection
	- variable mapping
	- splitting into basic blocks

- The major step being replaced with an LLM is the translation of a basic block.

- The translation at basic-block level, assuming local variables are managed on the stack (without dealing with register assignment) and swapping in/out from registers on every entry/exit to a statement is quite naive.

- the fact that you can translate the instructions of each basic block separately is the basis of traditional compilers. The more intricate challenges are around register allocation and memory allocation across blocks. Not to mention optimizations, as you declared these to be out of scope.

- In C.3 you compare the computation cost of LEGO-Compiler to the human effort require to build a compiler. Most of the human effort is outside what is done by the LEGO-compiler. The decomposition into basic blocks and non-optimized translation is achievable within undergraduate level compiler homework.

- Traditional compilers provide a deterministic and predictable outcome. The computational effort associated with your approach is orders of magnitude bigger, and provides no guarantees. Why would we take this approach?

- I would guess that there is exploring optimizations can be more fruitful for this technique.

(2) what are the assumptions you are making about the programming language and the translation?

Your translation is based on the fact that each statement is completely independent. The assumption of composability (Appendix B) does not hold when memory allocation is permitted (due to aliasing).

Consider the program:
```
int *p = malloc(sizeof(int));
int *q = p;
*p = 10;
*q = 20
```

The code is supposed to:

1.	Allocate memory for an int and store it in p.
2.	Make q point to the same memory as p.
3.	Store 10 in the memory pointed to by p.
4.	Store 20 in the same location via q, so the final value in memory is 20.

However, if translation treats *p = 10 and *q = 20 as independent, it could lead to the (incorrect) assumption that p and q point to different locations, and might (incorrectly) allocate separate memory for p and q.


Another (somewhat similar) issue may arise if the LLM decides to re-order instruction in a block without observing that they are dependent (for example, when operating on cells of an array, assigning them one after the other).




# Additional Questions


- You say "we don't rely on an existing compiler" = how do you get the examples for ICL? What is "compiler knowledge guided CoT"

- what does "trivial semantic units" mean?

- what do you mean by "semantically commposable control blocks", are these just basic blocks as implied later?

- what is the difference from how a traditional compiler works to "translate" each block separaetly, while maintaining a symbol table? What part of the compiler are you really replacing? The syntax-directed translation rules? What is the value in replacing these?

- who is responsible for allocating the addresses? For example, who assigns blksize to -36(%rbp) in Figure 1(b).


- Line 270-272, can you provide more details about this renaming pass? is it aware of scoping rules?

- Line 322-323 - 99% accuracy on a testset of 500 examples from ExeBench is very different from the "over 99% on ExeBench" (40k examples, or 23k examples after filtering). A different claim at a factor of x46.

# Minor

- l 62 - "we [do] not"

---

> ### Author Response · Authors · 2024-11-17
> **Response to Reviewer Mneu (Part 1 of 2)**
>
> Dear Reviewer Mneu:
>
> Thank you for your in-depth review. Please find the responses to the weaknesses and questions below:
> # 1. What is the motivation for this work?
> In short, our work is primarily a natural extension of neural code translation, with neural compilation being a special case. We first found that LLMs struggle to obtain very high compilation accuracy, so we experimented with various methods to improve this, and the methods (except the LEGO translation) are outcomes of these efforts. However, even with these improvements, we could not handle lengthy programs, even when the code consisted of simple repetitions, such as 100 for(i) loops handling simple statements. This motivated us to study the properties of programming languages. We also had an intuitive sense of the composability property we formalize, as this is the foundation of how compilers work.  Then, we realized that the translation process is composable. Unlike natural languages, code may be complex in design, but translating well-designed code is not necessarily difficult. As a result, we proposed both the LEGO translation methodology and the LEGO-Compiler system. The former is applicable not only to neural compilation but to a broader range of code translation tasks (and possibly more), while the latter is a significant step forward for neural compilation, as it is the first to successfully translate real-world codebases like CoreMark.
> # 2. 99% accuracy on a testset of 500 examples from ExeBench is very different from the "over 99% on ExeBench" (40k examples, or 23k examples after filtering). A different claim at a factor of x46.
> We apologize for the simplification here. However, we believe using a subset of ExeBench is already representative because we find the difficulty distribution and accuracy results similar across ExeBench. A medium-sized subset can represent ExeBench well and allows us to perform more experiments efficiently. Since testing the full 23k dataset using commercial LLMs would require significant time and financial resources, we chose the current 500-sample subset for experiments with more LLMs and to ablate our methods. We are confident that the results will be consistent between the subset and the full 23k ExeBench. We are performing full scale ExeBench evaluation and we will report the results later.
> # 3. What are the assumptions you are making about the programming language and the translation?
> We need to clarify that our LEGO-Compiler can handle the scenario you provided easily, as it compiles each line one by one. The correctness of the code itself (where aliasing complicates programming and optimization) is the user’s responsibility, and since our method currently does not target optimizations, the translation of cases with pointer aliasing will still work correctly. The assumptions we make to programming languages are stated in **Appendix B.5** and a few more in **Appendix E**.
> ```c
> int *p = malloc(sizeof(int));
> int *q = p;
> *p = 10;
> *q = 20
> ```
> The provided code example will translate to the following correct output by Claude 3.5, with similar outputs for other LLMs:
> ```x86
>     pushq    %rbp
>     movq    %rsp, %rbp
>     subq    $16, %rsp        # Space for two pointers p and q
>     # malloc call
>     movl    $4, %edi         # sizeof(int) = 4
>     call    malloc@PLT
>     # p = malloc result
>     movq    %rax, -8(%rbp)   # Store malloc result in p
>     # q = p
>     movq    -8(%rbp), %rax
>     movq    %rax, -16(%rbp)  # Store p in q
>     # *p = 10
>     movq    -8(%rbp), %rax
>     movl    $10, (%rax)      # Dereference p and store 10
>     # *q = 20
>     movq    -16(%rbp), %rax
>     movl    $20, (%rax)      # Dereference q and store 20
> ```
> # 4. How do you get the examples for ICL? What is "compiler knowledge guided CoT"?
> The 1-shot example is carefully crafted to demonstrate the principles of CoT and to align the output. “Compiler knowledge guided CoT”  refers to subtasks such as variable capturing, reasoning, renaming, stack allocation, etc. We call it compiler knowledge guided because we are inspired by compiler design experience. However, the CoT process itself does not depend on any specific compiler implementation, so it remains independent of existing compilers.

---

> ### Author Response · Authors · 2024-11-17
> **Response to Reviewer Mneu (Part 2 of 2)**
>
> # 5. What do you mean by "semantically composable control blocks"?
> We mean control block by full code patterns like if(){} and for(;;){}, etc, they are naturally encapsulated code patterns in modern PLs, and are LLM-aware since LLMs can reason their scope from bracket pairing, they are proved to be composable during translation, and they are a natural granularity level between strict basicblock and function. Function level neural translation is the majority and we think it is too coarse-grained for LLMs, making it hard to scale up. As for basicblocks defined in the compiler community, they are intuitively fine-grained and perform well in LEGO translation. However, we think control blocks are better semantic units as they match the full control keyword statements, naturally encapsulated code patterns, and LLMs are pretrained well to translate these control blocks.
> # 6. What is the difference from how a traditional compiler works to "translate" each block separately, while maintaining a symbol table? What part of the compiler are you really replacing? The syntax-directed translation rules? What is the value in replacing these?
> The translation of each block is inspired by how O0 compiler compiles, therefore, philosophically similar, the major differences are that we directly handle with source code and do the renaming, mapping, allocation ahead in the symbol table, which may differ from traditional way that usually operates at lower level like IRs or ASTs. We bypass the compiler part of compiler by using LLMs to translate from source code to assembly code. We treat neural compilation as a special case of neural code translation, therefore, both the AST building, traversing, syntax-directed translation are replaced because we fully rely on a LLM to compile/translate. The whole process is illustrated in **Figure 2**.
>
> The values of these processes are:
> - 1)We prove LLMs can follow carefully designed CoTs to do compiler tasks, which means the traditional compiler development knowledge can be taught to LLMs.
> - 2)We are the first to realize that **neural translation of function level code can be further split up** into finer granularities (control block, basic block, or even statement), because of the translation composability nature, and therefore, be capable of handling reasonably sized programs by translating them separately.
> - 3)We theoretically prove the translation composability, which is actually not a totally new theorem but an intuitive formalization of how compilers handle programs.
> # 7. Who is responsible for allocating the addresses? For example, who assigns blksize to -36(%rbp) in Figure 1(b).
> LLMs allocate the addresses, it is a reasoning process to sequentially allocate the stack addresses from the first variables to the last, it is majorly integer add/sub/mul mathematical computations to calculate an offset, and we found LLMs can handle them effectively as the CoT approach outperforms direct generation approach. The details of such process is illustrated in the Variable Mapping part in **Figure 2**.
>
> Additionally, In our implementation, the whole CoT process is done by LLM, however, we agree that either the frontend part to form the translation rules and intermediate results or the backend part that do the translation of each part can be done by traditional methods, so there are exploration space for more suitable designs. Currently we implement a LLM-only approach, majorly to showcase LLMs’ capabilities in complex code translation tasks(compilation). We believe we study the hardest neural code translation tasks to the best of our knowledge.
> # 8. Line 270-272, can you provide more details about this renaming pass? Is it aware of scoping rules?
> The renaming pass first needs another analyzer pass that prompts LLMs to reason about every variable with their living scope. For scopes in C, LLMs majorly need to reason about every variable about its definition, since every variable has its own living scope, and the innermost definition will mask the rest. Therefore, for each identifier, the LLMs match the closest definition (from the current depth, cascading out to the global scope, similar to symbol resolution in compilers).
> Later after analysis, we rename them, eliminating duplicate names so every entry in the symbol table (shown on the upper right side in **Figure 2**) will be unique for later allocation. This process is done purely at the source code level, and therefore, **can be easily checked** for equivalence through unit tests.
>
> It is aware of scoping rules. These rules are generally pretrained in LLMs, as we have found that LLMs can handle most renaming cases correctly in 0-shot. We further consolidate their understanding of scoping rules by providing a 1-shot ICL example.
>
> We hope the above responses help address your concerns and give you a clear impression. Thank you again for your valuable reviews!
>
> Sincerely,
>
> Submission931 Authors

---

> > ### Comment · Reviewer_Mneu · 2024-11-17
> >
> > Thank you for the clarifications.
> >
> > # 6 Block translation
> >
> > > 2)We are the first to realize that neural translation of function level code can be further split up into finer granularities (control
> > > block, basic block, or even statement), because of the translation composability nature, and therefore, be capable of handling
> > > reasonably sized programs by translating them separately.
> >
> > Translation, whether neural or otherwise, is known to be applicable at these levels. The concepts of basic blocks, and definitely of syntax-directed blocks are inherent in how compilers work.
> >
> > > 3)We theoretically prove the translation composability, which is actually not a totally new theorem but an intuitive formalization > of how compilers handle programs.
> >
> > I agree on both counts. This is not a totally new result, and it is how compilers handle programs.
> >
> > # 7 Allocating the addresses
> >
> > Mixing neural and classical approaches for this makes sense.
> >
> > # 8 Renaming pass
> >
> > Thanks, this helps.

---

> > > ### Author Response · Authors · 2024-11-20
> > > **Further discussions with Reviewer Mneu**
> > >
> > > Thank you for your precious time investigating our responses. We will continue to discuss the remaining concerns.
> > > ###  Translation, whether neural or otherwise, is known to be applicable at these levels. The concepts of basic blocks, and definitely of syntax-directed blocks are inherent in how compilers work.
> > > We acknowledge that for traditional non-neural translation methods, like compilers or transpilers, block level translation is intuitive and doesn’t need to be emphasized. However, to the best of our knowledge, our approach is the first to demonstrate that neural translation can effectively operate at various sub-function granularities while maintaining translation quality, which enables handling of larger programs that were previously intractable for end-to-end neural methods. Note that it requires special non-trivial handling of context, segments to do so with LLMs, and we use the neural compilation part as an illustration for this. This bridge between traditional compiler wisdom and neural approaches opens new possibilities for scaling neural code translation to practical applications.
> > >
> > > ### I agree on both counts. This is not a totally new result, and it is how compilers handle programs.
> > > We appreciate the reviewer's agreement on how the proof should be evaluated and we would like to supplement why we need such a proof. The motivation for providing this formal proof stems from a significant gap in the literature: despite these properties being fundamental to compiler design, we could not find any scientific papers in the past three decades that formally establish these properties in a way that can directly apply to neural translation methods. While related concepts exist in work like incremental compilation[1] or general compiler educational textbook[2], they address different technical challenges or cover more general aspects in compiler designs. Technically, the block level translation composability is not well formalized.
> > >
> > > Our formalization provides a formal proof of the composability nature during code translation, bridges this temporal and conceptual gap between traditional compiler theory and modern neural methods, and provides a detailed example case of neural compilation to guide future research.We think it is valuable as neural approaches to code translation become more prevalent in the field.
> > >
> > > [1] Steven P. Reiss. 1984. An approach to incremental compilation. SIGPLAN Not. 19, 6 (June 1984), 144–156. https://doi.org/10.1145/502949.502889
> > >
> > > [2] Wirth, Niklaus, et al. Compiler construction. Vol. 1. Reading: Addison-Wesley, 1996.
> > >
> > > We hope the above discussions address your concerns. Please let us know if there are other concerns related, typically these concerns we clarified earlier in Response (Part 1 of 2).

---

> ### Author Response · Authors · 2024-11-22
> **Update with new evaluation results**
>
> Dear reviewer Mneu:
>
> We have just updated our pdf version with new evaluations showcasing the effectiveness of LEGO translation method. Besides, we also perform a 10x larger scale evaluation on ExeBench with Deekseek model, where the results are as expected, consistent between the 500 subset and the newly evaluated 5000 subset. Due to time constraint we cannot perform full scale 23k ExeBench evaluation, however, we can make this update in later times after the rebuttal period.
>
> We hope the above updates address your concerns.
> **If our answers are more in line with your expectations, we kindly invite you to reconsider your initial rating.**
>
> Sincerely,
>
> Submission931 authors

---

> ### Author Response · Authors · 2024-11-30
>
> Dear reviewer Mneu:
>
> We would like to remind you that the discussion period is ending, and we haven’t heard from your responses where we have added several important experiments and have already addressed reviewer wpuL’s concerns.
>
> We would really appreciate it if you could spent some time to review the rebuttal we have updated and update your review. We will be happy to clarify any questions related.
>
> Sincerely,
>
> Submission931 authors

---

### Official Review · Reviewer_KZKL · 2024-11-04

**Soundness:** 2
**Presentation:** 3
**Contribution:** 3
**Rating:** 6
**Confidence:** 2

**Summary:**

The paper presents a novel neural compilation system aimed at translating high-level programming languages into assembly code using Large Language Models (LLMs). Recognizing LLMs' limitations in handling complex and lengthy code, the authors introduce "LEGO translation," which decomposes large programs into smaller, manageable blocks (akin to LEGO pieces) that can be individually translated and reassembled. This process is enhanced by a Chain-of-Thought (CoT) approach and a feedback-driven self-correction mechanism, helping LLMs manage the intricate steps of code compilation more accurately.

Contributions include:

1. **LEGO Translation**: A modular approach to breaking down programs into composable blocks, making translation more efficient and scalable.
2. **Annotation-Based Chain-of-Thought (CoT)**: LLMs are guided through the compilation process with intermediate annotations, improving context comprehension and translation accuracy.
3. **Feedback Mechanism**: A self-correcting system that uses error feedback for iterative improvement in the translation process.
4. **Empirical and Theoretical Validation**: LEGO-Compiler achieves over 99% accuracy on ExeBench and fully compiles CoreMark, demonstrating its potential for handling code complexity and scalability.

**Strengths:**

1. **Originality**: The paper introduces several novel concepts, primarily the "LEGO translation" approach. This method effectively decomposes complex code into smaller, semantically composable blocks, allowing for more scalable and accurate translations. This compositional approach is particularly original in the neural compilation field, where such modular treatment of code is uncommon. Furthermore, the annotation-based Chain-of-Thought (CoT) and feedback-driven self-correction mechanisms are innovative enhancements that showcase a creative blend of ideas from traditional compiler techniques with modern neural architecture capabilities. By positioning LEGO-Compiler as a system capable of complementing traditional compilers, the paper broadens the scope of neural compilation research.

2. **Quality**: The paper provides a robust empirical validation of the LEGO-Compiler’s effectiveness across diverse benchmarks, such as ExeBench, CoreMark, and the newly introduced LongFunction dataset. The results are impressive, showing over 99% accuracy on ExeBench and full compilation success on CoreMark, indicating the system's capability to handle real-world, industry-grade code. The inclusion of both theoretical and empirical support, including formal proofs of composability, reinforces the quality and rigor of the research.

3. **Clarity**: The paper is well-structured and clearly explains the methods used, such as the part-split algorithm, control flow annotations, struct annotations, and feedback mechanisms. Visual aids, like flowcharts and annotated code examples, further enhance understanding, particularly for complex concepts like variable mapping and CoT stages. Overall, the clear presentation makes the LEGO translation process and associated mechanisms accessible to readers with varying levels of expertise in neural compilation and compiler design.

4. **Significance**: This work makes a strong case for the potential of LLMs in neural compilation, an area traditionally dominated by classical compilers. By demonstrating that LLMs can efficiently handle code translation tasks that were previously infeasible due to context limitations, the paper sets a foundation for future research on LLM-based compilation. The proposed modular approach to code translation may influence other domains where handling long or complex code sequences is challenging. Additionally, the introduction of the LongFunction dataset provides a valuable benchmark for evaluating neural compilers, potentially serving as a standard for future advancements in the field.

In summary, the paper’s originality, comprehensive evaluation, clear articulation, and impactful contributions make it a substantial addition to the literature on neural compilation. The LEGO-Compiler represents a significant step forward in addressing scalability issues and extending the capabilities of LLMs in code translation, providing a solid framework for further research and practical applications.

**Weaknesses:**

**Weaknesses**

While the LEGO-Compiler paper makes substantial contributions to the field of neural compilation, there are a few areas where improvements could enhance its impact and clarity. I offer constructive feedback on the following points to help the work align more closely with its goals:

1. **Scalability and Efficiency**: Although the paper demonstrates the scalability of LEGO-Compiler on long functions and complex codebases like CoreMark, it acknowledges higher computational costs relative to traditional compilers. However, the paper stops short of providing any concrete analysis of these costs or suggesting optimizations to mitigate them. Including a quantitative comparison of LEGO-Compiler's efficiency against traditional methods (e.g., runtime, memory usage) would provide a clearer understanding of its real-world applicability. Additionally, further exploration of methods to reduce computational demands, such as model pruning or more efficient context handling within the LLM, could add significant value.

2. **Evaluation Breadth and Context**: While the empirical results on ExeBench and CoreMark are compelling, the choice of datasets might limit the generalizability of the findings. ExeBench and CoreMark focus on specific C code structures, leaving open questions about how LEGO-Compiler would perform on code from other languages or highly varied program structures. An evaluation across different programming languages, such as Java or Python, would highlight the flexibility of LEGO-Compiler and make the case for its broader applicability. Furthermore, testing on real-world projects with interdependencies, libraries, or complex input-output structures would more fully reflect the challenges neural compilers face outside of controlled benchmarks.

3. **Error Feedback and Self-Correction Limitations**: Although the paper proposes a feedback-driven self-correction mechanism, it appears to rely on error feedback types that may not always be straightforward to integrate in practice (e.g., assembler, runtime, and behavioral error feedback). Each type of error feedback requires specific handling, which could complicate implementation in a more generalized setting. It would be beneficial if the paper discussed how to automate or streamline these correction processes more effectively. Providing more specific insights or examples on how this system could handle unforeseen or non-standard errors could also strengthen the approach and demonstrate the robustness of LEGO-Compiler’s feedback mechanism.

4. **Detailed Ablation of LEGO Translation**: While the LEGO translation method is central to the paper's contributions, the empirical results could benefit from a more detailed ablation study focused specifically on this component. For example, examining how various part sizes or splitting strategies affect accuracy and performance on different codebases would provide insights into the optimal configurations of LEGO translation. This could be helpful for future researchers or practitioners looking to adapt or refine the approach.

5. **Limited Discussion of Edge Cases and Limitations**: Although LEGO-Compiler achieves high accuracy on ExeBench and CoreMark, the paper does not explicitly discuss any scenarios where the system may fail or struggle. For example, specific cases of nested or non-linear control flows, rare syntax patterns, or highly recursive functions could challenge the system’s modular translation approach. Highlighting these limitations, possibly with failure case analyses or suggestions for future handling of edge cases, would provide a more balanced perspective on LEGO-Compiler’s strengths and limitations.

6. **Potential for Fine-Tuning or Model Customization**: The paper could discuss in greater depth how LEGO-Compiler’s performance might benefit from fine-tuning on specific code domains or architectures, particularly given the reliance on LLMs trained on general datasets. Tailoring the model to domain-specific code (e.g., embedded systems, high-performance computing) could optimize its accuracy and efficiency for these applications. The authors could address whether or not such fine-tuning might enhance LEGO-Compiler’s flexibility and discuss how researchers might incorporate this approach in practice.

In summary, while LEGO-Compiler shows strong potential, enhancing the analysis of its scalability, generalizability, error-handling robustness, and application-specific fine-tuning could provide a more comprehensive view of its capabilities and limitations. These adjustments would strengthen its contribution to neural compilation and broaden its applicability to diverse real-world scenarios.

**Questions:**

1. **Computational Costs and Efficiency**: Can the authors provide a more detailed comparison of LEGO-Compiler’s computational efficiency relative to traditional compilers? Specifically, it would be helpful to know if there are areas where LEGO-Compiler could optimize computational resource usage or if there are plans to explore methods, like model pruning or improved in-context memory, to enhance efficiency without sacrificing accuracy.

2. **Applicability Across Languages**: Given that the experiments focus primarily on C code, how adaptable is LEGO-Compiler to other programming languages, especially those with different memory management paradigms (e.g., Java) or highly dynamic types (e.g., Python)? Could the authors clarify any plans for testing the approach on different languages and any anticipated challenges they foresee in these cases?

3. **Error Feedback Mechanism**: The feedback-driven self-correction mechanism relies on different feedback types (assembler, runtime, and behavioral). Could the authors elaborate on how feasible it is to implement each type of feedback in practice and discuss any automation strategies they envision to simplify the process? Additionally, are there specific cases or error types where this feedback approach might struggle?

4. **Granularity of LEGO Translation**: The paper introduces LEGO translation as a modular, part-split approach to handling complex code but lacks detailed ablation on part size selection and splitting strategies. Could the authors share insights or planned experiments on how they arrived at optimal part sizes? For instance, have they explored whether finer or coarser splits might impact translation accuracy differently across datasets or LLM models?

5. **Handling of Edge Cases and Recursion**: Does LEGO-Compiler currently have specific strategies for handling edge cases such as deeply nested loops, irregular control flows, or highly recursive functions, which may challenge modular composition? If so, could the authors elaborate on these strategies? If not, do they have ideas for how these cases could be integrated into the current approach?

6. **Potential for Model Fine-Tuning**: Given that LEGO-Compiler leverages general LLMs, would the authors consider fine-tuning models on specific types of code (e.g., embedded systems, HPC) to improve results further? Could they clarify if they have experimented with or plan to experiment with domain-specific tuning, and if so, any notable improvements observed?

7. **Comparison with Non-LLM Neural Compilers**: How does LEGO-Compiler compare to other non-LLM-based neural compilers or neural translation systems, both in terms of accuracy and scalability? Including this comparison, especially if backed by experimental data, would strengthen the paper’s case for adopting LLMs in neural compilation. Could the authors provide clarification or consider adding this in future revisions?

---

> ### Author Response · Authors · 2024-11-17
> **Response to reviewer KZKL (Part 1 of 2)**
>
> Dear Reviewer KZKL:
>
> We sincerely thank you for your thorough and insightful review of our paper. Your comprehensive summary and analysis demonstrate a precise understanding of LEGO-Compiler's core contributions. We are particularly impressed by your accurate articulation of our key innovations: the LEGO translation approach—a novel method that decomposes complex code into semantically composable blocks—and the annotation-based Chain-of-Thought mechanism, which, for the first time, introduces reasoning capabilities into neural compilation. Furthermore, to the best of our knowledge, LEGO-Compiler is the first work to enable LLMs to handle real-world, industry-grade code translation. These techniques are not only applicable to neural compilation but are also relevant to broader code translation tasks, particularly the LEGO translation method.
>
> We note that while you've given our paper a strong accept rating (10), you've indicated a lower confidence level (2) in your review. We believe we can address this confidence gap, as your detailed comments demonstrate a solid grasp of our paper's central contributions and technical details, and a clear understanding of where our novelty lies. Your questions and constructive feedback are precise and well-targeted, suggesting a better understanding than you might have credited yourself with. We will address each concern:
> # 1. Computational Efficiency and Scalability
> Although our goal is not to replace traditional compilers, knowing its translation cost is indeed important. We will make a qualitative discussion here and later revise it in our manuscript. Using a 200 token program as example, GCC compiles in ~20ms, while generating 600 token assembly output for a LLM, say Llama-3.1-8B serving with vllm, its Time-Per-Output-Token is roughly 40ms/token for one A100 80GB GPU, therefore, will take 20 second for output, which is 2-3 orders of magnitude slower, using CoT will further increase the token output and therefore, being slower. This does show that using LLM to compile is costly compared to using modern compilers, however, as other machine translation tasks are also costly using decoder-only LLMs, we cannot blame them because knowing the translation capability with LLMs is also important.
> As for resource efficiency optimization, since LEGO translation is modular and can be translated separately, there could be optimizations by serving independent translation steps into different LLM chats, which could batch the translation steps together and speed up the generation. Functions can naturally be parallelized, and in our work, we further break down tasks into control blocks. However, this may require the frontend analyzer to be more carefully prompted or non-LLM implemented to support robust finer-grained translation parallelism. We believe this could be an interesting topic for future work.
> # 2. Evaluation Breadth and Context
> Currently we evaluate our LEGO-Compiler on C programming languages, and we also discuss the potential with other programming languages in **Appendix E**, the limitation section. There will be new challenges for different programming languages, however, we believe using similar CoTs and the LEGO translation method will definitely be helpful. We target more detailed study on different language compilation and translation as future work.

---

> ### Author Response · Authors · 2024-11-17
> **Response to reviewer KZKL (Part 2 of 2)**
>
> # 3. Error Feedback and Self-Correction Limitations
> Yes, the error feedback surely needs some development to support LLMs to do so, but not much, we will discuss the details: The assembler error can be easily captured by assembler toolchain and is the best self-correction type. As for runtime errors (usually a core dump), we need to use assembler to assemble with debug info(-g) to get the executable and capture its traceback from gdb execution, which can also help locate a range of possible errors. The silent error(mismatch output) is the one that LLMs are more struggling with, as even for humans, identifying silent errors is also challenging. Possible solutions include using binary search recursively in the code, and LLM simulating execution with the code, to locate the error position. This is an interesting topic waiting for further investigation.
> # 4. Ablation of LEGO Translation Granularity
> We use an adaptive algorithm (**Algorithm 1**) with error feedback to decide a suitable size for LLM to translate. From our investigation, translation with finer granularity will help the translation accuracy as it lowers the difficulty of each translation step, and it is the reason for us to use an adaptive way, from coarse-grained functions(no split) to fine-grained blocks or even statements(current split limitation).
> # 5. Limited Discussion of Edge Cases and Limitations
> Please refer to **Appendix E** for detailed limitation discussions. Besides, we will soon add more edge case failure examples to the manuscript to showcase the boundary of current methods with LLMs for clarity.
> # 6. Potential for Fine-Tuning or Model Customization
> Yes, compared to using ICL, fine-tuning the model with a compilation dataset should boost the results by learning the translation into parameters when performing direct translation. As it is already reported by [1], which is the SOTA in neural compilation task before our method. However, with only fine-tuning in raw dataset may not boost its accuracy in our LEGO-Compiler method as the instruction-following ability of the model may decrease during the fine-tuning process. Actually we prefer another perspective, where we don’t rely on existing compilers, we actually let LLMs reason the translation process and perform the translation as a “compiler”. Ultimately, we believe LLMs can reason the translation for other languages and architectures, as long as their monolingual corpora are pretrained, and few or no aligned corpora would be required. However, this is just for discussions and needs future work to further prove its correctness.
>
> [1] *Introducing Compiler Semantics into Large Language Models as Programming Language Translators: A Case Study of C to x86 Assembly (Zhang et al., EMNLP Findings 2024)*
> # 7. Comparison with Non-LLM Neural Compilers
> Please refer to our **3rd response in the general reply**, where smaller non-LLM neural compilers are performing worse than foundation LLMs, therefore we think it’s not necessary to compare with.
>
> We are sorry the response is a bit lengthy. We hope these clarifications not only address your concerns but also give you greater **confidence** in your assessment of our paper's contributions to the field.
>
> Sincerely,
> Submission931 Authors

---

> ### Author Response · Authors · 2024-11-22
> **Update with new evaluation results**
>
> Dear Reviewer KZKL:
>
> We have currently updated our manuscript and conducted both more real-world C program evaluation and randomly generated program evaluation, with method ablations. We have updated the results in Appendix D.4 and Appendix D.5, where the results do show the effectiveness of LEGO translation method. We also perform larger scale evaluation on ExeBench which shows consistent results.
>
> Additionally, we find more edge cases and error types through the newly conducted evaluations, we discuss them in Appendix D.4.
>
> We hope you find these newly updated results interesting and that **they strengthen your confidence** in your assessment of our work!
>
> Best regards,
>
> Submission931 authors

---

> ### Author Response · Authors · 2024-11-30
>
> Dear reviewer KZKL:
>
> Thank you again for your great review, and we would like to remind you that the discussion period is ending, and we haven’t heard from your responses where we have added several important experiments and have already addressed reviewer wpuL’s concerns.
>
> We would really appreciate it if you could spent some time to review the rebuttal we have updated and consider raising your **confidence score** if more details are explained clearly from our clarifications. We will be happy to clarify any questions related.
>
> Sincerely,
>
> Submission931 authors

---

### Author Response · Authors · 2024-11-22
**New updates**

Dear all reviewers,

We appreciate your valuable feedback and have updated our manuscript to address your concerns. Below, we outline the modifications made, with all changes highlighted in brown within the manuscript:

### Large-Scale Experiments:
In response to Reviewer Mneu's suggestion, we conducted additional large-scale experiments evaluating the DeepseekCoder model on a 10x larger randomly sampled subset of 5,000 cases from ExeBench (up from the previous 500 cases). While we were unable to process the full set of 23,000 filtered cases within the limited rebuttal time frame, we have updated Table 1 with the new results and provided explanations. Notably, these results are consistent with those obtained from the smaller subset, and therefore, supporting the representativeness of the results with other LLMs to represent ExeBench.
### Evaluation on Real-World Codebases:
Following the recommendations of Reviewers wpuL and vsfE, we expanded our evaluation to include more real-world codebases. We utilized AnsiBench (https://github.com/nfinit/ansibench), which comprises several well-known ANSI C standard benchmark suites, including Whetstone, Dhrystone, Linpack, Stream, and CoreMark. We performed ablation studies comparing baseline translation methods, translation with Chain of Thought (CoT), and LEGO translation on AnsiBench. The updated results can be found in Appendix D.4. Our findings demonstrate the effectiveness of LEGO translation; notably, it successfully processed a significant portion of complex code, passing 94 out of 96 cases. These results are visualized in Figure 8.
### Randomly Generated Programs:
In line with Reviewer wpuL's suggestion, we also evaluated randomly generated programs with sufficient complexity to assess the effectiveness of the LEGO translation method. We employed Csmith, a widely used random generator for C programs known for identifying compiler bugs. The evaluation results on Csmith are included in Appendix D.5. Given that Csmith-generated code does not inherently check for behavioral equivalence, we undertook additional modifications to facilitate testing. Although our current examination is on a smaller scale, it highlights significant advantages of using the LEGO translation method for compilation, as illustrated in Figure 9.
### Discussion of Limitations and Failure Cases:
We have expanded our discussion regarding limitations and included an analysis of failure cases to provide a more comprehensive understanding of our approach.
### Corrections:
Lastly, we have addressed minor typographical errors pointed out by reviewers throughout the manuscript.

Thank you for your constructive feedback; we believe these updates significantly enhance our work. Looking forward to your responses.

Sincerely,

Submission931 authors

---

### Author Response · Authors · 2024-12-04

We thank all the reviewers for providing constructive review of our paper. We appreciate the time spent to analyze our paper, specifically reading through the appendix section about the proof details. We appreciate the reviewers’ enthusiasm for our novel approaches regarding neural compilation (*“effectively decomposes complex code into smaller, semantically composable blocks, allowing for more scalable and accurate translations, modular treatment of code is uncommon”* by **KZKL**, *“The first work I'm aware of to show effective translation of reasonably sized programs, compelling approach”* by **Mneu**, *“Novel method that adopts the idea of decomposing and composing to handle large complex programs”* by **wpuL**). The proof of translation composability also strengthens the correctness of LEGO translation acknowledged by Reviewer **wpuL**, **KZKL** and **Mneu**(during rebuttal). Our experimental results of over 99\% accuracy are also convincing and interesting (*“Significance, a significant step forward”*, **KZKL**, *“hard and challenging problem, achieving accuracy reaching 99\%, interesting results”*, **vsfE**, *“evaluation results look promising, analysis of the failure cases is insightful and convincing”*, **wpuL** during rebuttal).

To address the reviewers’ concerns, we upload a new version of the paper, with the following substantial updates colored in brown color for clearance:

**Table 1**: We updated results with 10x larger scale evaluation on ExeBench to support our claim of results consistency and representativeness in ExeBench; Additionally, we compare with current SOTA work in neural compilation, showcasing our advancement in this field.

**Appendix D.4**: We expand more real-world codebases besides CoreMark, and showcase the significance of LEGO translation approach.

**Appendix D.5**: We add extra evaluation on random programs generated by Csmith, which also showcase the significance of LEGO translation.

Besides, other conceptual concerns raised by reviewers have been clarified during the rebuttal, the important ones are:

**Benchmark selection**, where CoreMark is not a toy benchmark, and ExeBench evaluation is also aligned with prior works.

**Language translation assumption**, where aliasing issues(by Mneu) and library usage concerns(by wpuL) are clearly explained with examples. They are not limitations of our method.

**Method significance**: we explain the accuracy metric can be viewed in another perspective, **how complicated cases can LEGO-Compiler translate correctly**, where improvement of the last few digits to 100% represents significant advancement. Additionally we also add extra evaluations to support the significance of LEGO translation, which does scale up by an order of magnitude.

We respond in detail to all the reviewers and will be happy to address any concerns the reviewers have during the discussion period.

---

### Meta-Review · Area_Chair_H7mw · 2024-12-22

**Metareview:**

This paper studies an important problem: using LLMs to compile programs. This paper proposes to use LLMs to compile programs like LEGO, i.e., breaking down the program into small pieces, compiling them individually with LLMs, and finally composing the compilation results through dedicated rules. Empirically, the proposed method achieves decent accuracy in determining the correctness of compiled programs. The paper evaluates its effectiveness across various synthesized and real-world programs. Except for programs generated by Csmith, the proposed approach achieves high accuracy.

This is a borderline paper. After extensive discussions, the AC recommends rejection because: (1) the technical innovation is limited; (2) the empirical results are not impressive enough.

**Additional Comments On Reviewer Discussion:**

There were discussions on the technical novelties and empirical results. The AC eventually decided to recommend rejection because technical novelties and empirical results do not reach the bar.

---

### Decision · Program_Chairs · 2025-01-22

Reject